# Environmental conditions associated with initial northern expansion of anatomically modern humans

**Frédérik Saltré** [1,2] ✉, **Joël Chadœuf**[3], **Thomas Higham** [4,5], **Monty Ochocki**[4,5], **Sebastián Block**[6], **Ellyse Bunney**[7], **Bastien Llamas** [2,8,9,10] & **Corey J. A. Bradshaw** [1,2]

The ability of our ancestors to switch food sources and to migrate to more favourable environments enabled the rapid global expansion of anatomically modern humans beyond Africa as early as 120,000 years ago. Whether this versatility was largely the result of environmentally determined processes or was instead dominated by cultural drivers, social structures, and interactions among different groups, is unclear. We develop a statistical approach that combines both archaeological and genetic data to infer the more-likely initial expansion routes in northern Eurasia and the Americas. We then quantify the main differences in past environmental conditions between the more-likely routes and other potential (less-likely) routes of expansion. We establish that, even though cultural drivers remain plausible at finer scales, the emergent migration corridors were predominantly constrained by a combination of regional environmental conditions, including the presence of a forest-grassland ecotone, changes in temperature and precipitation, and proximity to rivers.

The environmental mechanisms that shaped human cultural adaptation since the first anatomically modern humans dispersed out of Africa are still contested[1]. Human evolution and expansion over the last 120,000 years are hypothesized to have been environmentally determined[2] because the occurrence of orbital-scale climate shifts[3–5] might have created suitable habitat corridors for humans to exit Africa. However, cultural drivers, social structure, and interactions among different groups could have also explained movement patterns[6].

Identifying the relative role of ecological and cultural processes underlying the earliest events of human expansion requires accurate dating of the main out-of-Africa dispersal event(s), and a detailed quantification of subsequent expansions across other continents. This is challenging, because it is likely that there was more than one movement or dispersal event by early *Homo sapiens* out of Africa as early as ~177 ka (1 ka = 1000 years ago) in south-western Asia[7], ~100–80 ka in China[8], and ~210 ka in Europe[9]. The general consensus is therefore that humans exited Africa sometime between these early dates and ~65 ka[10], eventually to people the rest of the world. Recent evidence suggests that humans were present in areas of southern Europe by 46 ka[11,12] and high latitudes by 32 ka[13–15] before reaching

[1]Global Ecology | Partuyarta Ngadluku Wardli Kuu, College of Science and Engineering, Flinders University, GPO Box 2100 Adelaide, SA 5001, Australia. [2]Australian Research Council Centre of Excellence for Australian Biodiversity and Heritage, Wollongong, NSW, Australia. [3]UR 1052, French National Institute for Agricultural Research (INRA), Montfavet, France. [4]Department of Evolutionary Biology, University of Vienna, University Biology Building, Carl Djerassi Platz 1, A-1030 Wien, Austria. [5]Oxford Radiocarbon Accelerator Unit, Research Laboratory for Archaeology and the History of Art, University of Oxford, Oxford OX1 3TG, UK. [6]Department of Ecology and Evolutionary Biology, Princeton University, Princeton, NJ 08544-1003, USA. [7]Commonwealth Scientific and Industrial Research Organisation, Urrbrae, SA 5064, Australia. [8]Australian Centre for Ancient DNA, School of Biological Sciences, The Environment Institute, University of Adelaide, Adelaide, SA 5005, Australia. [9]National Centre for Indigenous Genomics, John Curtin School of Medical Research, Australian National University, Canberra, ACT 2601, Australia. [10]Indigenous Genomics, Telethon Kids Institute, Adelaide, SA 5000, Australia. ✉ e-mail: frederik.saltre@flinders.edu.au

Beringia[16]. The long-delayed entry into North America by ~16 ka[17], and then a quick expansion as far as southern South America by at least ~14.6 ka[18], do not necessarily preclude the proposal that some small groups could have entered North America before the Last Glacial Maximum (26.5–19 ka)[19,20].

When focussing on how past environmental changes might explain this general pattern of human expansion, many climate-driven hypotheses are at the core of the plausible mechanisms, such as the return of moist and mild environmental conditions in Europe arising from the resumption of the North Atlantic meridional overturning circulation[21], or a tropical-like climate across southern Asia[22]. However, in addition to shifts in temperature and other physical barriers (e.g., massive glaciers, high-elevation mountain ranges, etc.) impeding human migration[23,24], access to freshwater sources (rainfall driven), and the type of ecosystems encountered are hypothesized to have constrained human dispersal. This is because access to water could have physiologically constrained people to follow river courses (and only occasionally, coastlines)[25], with the optimal ecological niche characterized by a mix of open, savanna-type woodlands, wetlands, and rocky habitats[26].

Previous studies investigating human expansion have thus far relied either on archaeological evidence[20,27–29] or mitochondrial/whole-genome data[11,13,30]. However, these data are often sparse and incomplete, and population turnover can erase ancestral genetic signals. We must also consider the possibility that the earliest settlers might not have contributed to the gene pool[31,32], thus making the reconstruction of regional patterns of human expansion challenging when independent datasets are viewed in isolation. The lack of spatially and temporally continuous data has restricted testing hypotheses regarding environmental drivers of human expansion to qualitative scenarios that are often difficult to apply at continental scales[27,33,34]. The main problem is that even reliably dated and comprehensive archaeological evidence is unlikely to reflect the true timing of human arrival in an area due to a bias introduced by incomplete sampling or taphonomy[35].

Modelling approaches have been applied at coarse spatial scales to explore the realism of past human-expansion patterns across various regions of the globe[3,36,37], but such models rely on untested assumptions regarding ancient human demography included a priori in the model design[3,36]. In addition, the robustness of ensuing model predictions is evaluated by matching outputs either to the archaeological record[3] or genetic data[11], but these datasets are individually too sparse to construct continuous spatiotemporal maps of the regional patterns of expansion for comparison. Fortunately, recent statistical approaches have been developed recently to enable mapping of spatially continuous estimates of disappearances or appearances from the landscape based on palaeontological and archaeological records simultaneously[38].

Here, we developed and applied a new version of this statistical approach that overcomes methodological limitations to available radiocarbon data showing the presence of humans in Eurasia and the Americas after they exited Africa over 65 ka[10]. We then combined the archaeologically derived maps with the spatial patterns of genetic difference in human populations based on present-day human mitochondrial DNA to test the hypothesis that if environmental suitability was a determinant of initial expansion, then the preferential routes travelled by humans followed distinct combinations of environmental conditions. Specifically, we expected the routes to occur in regions with mid-range temperature and precipitation conditions[2,25], ecotones between forests and open habitats, based on the evidence of higher biomass at mid-range productivity[39], low landscape ruggedness to facilitate movement[23], and in regions with ample water supply (e.g., nearest rivers) and/or close to coastlines to maximize the potential diversity of available water and food sources[40]. To test these hypotheses, we generated maps of continuous regional timing of initial human appearance based simultaneously on both archaeological

records and genetic divergence data. We applied a customized pathway algorithm to these two maps to calculate a set of more-likely routes of initial peopling across the continents. We then quantified the difference in environmental conditions between the more-likely routes and other potential (less-likely) routes of expansion. We show that changes in forest cover, temperature, and precipitation shaped regional pattern of human migrations. Although their relative importance varied by region, humans predominantly travelled along routes that were warmer, more humid, and often near forest-grassland ecotones. We also find that river proximity influenced local migration patterns, especially in areas experiencing severe climatic conditions. We conclude that spatial scale (e.g., local, regional, continental) dictated the degree to which environmental determinism explains ancient human behaviours.

## Results and discussion

### Expansion routes

We calculated the more-likely routes travelled across northern Eurasia and the Americas between sets of two locations (i.e., a source and a destination), each representing the extremes of plausible pathways (brown-yellow markers in Fig. 1). These travelled pathways are based on two independent proxies: (i) continuous maps of estimated regional timing of arrival generated from archaeological records (Supplementary Fig. 1a and Supplementary Data 1), and (ii) maps of mitochondrial DNA-based fixation index ($F_{ST}$) measuring genetic distance between point estimates (Supplementary Fig. 1b). We reconstructed the geographic patterns of the timing of human expansion by applying a comprehensive, spatio-temporal method (see Methods and ref. 38) to the high-quality (i.e., suitable dates; Supplementary Data 1) archaeological records scattered across Eurasia and the Americas (Supplementary Fig. 1a, see Supplementary Discussion 1). We extracted the genetic dataset from a GenBank search query to find 27,506 human mitochondrial control-region sequences with reliable geographical locations, cleaned of ancient DNA data and any non-Indigenous lineages. We used a modified, least-cost path algorithm to calculate the likelihood of each plausible route being the more-likely travelled between each pair of source-destination locations (Supplementary Fig. 5). Here, we optimized the selection of each grid cell across the landscape to build a path between a source and its destination as a function of the lowest cost (see Methods). We calculated this cost in two different ways: (i) as the difference in timing of initial arrival between neighbouring grid cells (Supplementary Fig. 1a), and (ii) as the pairwise-distance of mitochondrial $F_{ST}$ (Supplementary Fig. 1b) between neighbouring grid cells. Optimizing for the lowest cost provides both the smallest difference in timing of arrival and the shortest genetic distance between all grid cells along the pathway (Supplementary Fig. 5). We modified this algorithm by adding a stochastic, spatial-decision process to account for exploratory behaviour by humans.

The more-likely routes derived from the two datasets indicated human spread across south-western Europe from Southwest Asia via the Caucasus Mountains toward Scandinavia (~48.3 ka) and moving around the Black Sea eventually to enter western Europe from north of the Alps[1]. Another route into Europe emerged via the Dalmatian and Mediterranean coasts (~44.1 ka; Fig. 1, and Supplementary Fig. 5). The route southeast to the Caspian Sea (Fig. 1) appears to be a critical area for human expansion because this was the main expansion pathway toward both Japan (~44.2 ka) and Beringia (~34.7 ka; Fig. 1). Our results show that the 'northern route' split to the east of the Caspian Sea, with a northern section skirting the Hindu Kush and the Pamir ranges[41,42] before moving into Mongolia (~47.1 ka), whereas the southern branch moved south of the Karakorum and Himalayan ranges (~45.8 ka)[28]. We acknowledge that our focus on the northern route ignores a potential southern route out of Africa to Asia through Ethiopia near the Red Sea[43] and the Arabian Peninsula[44], from where

humans could have spread rapidly into regions of Southeast Asia and Oceania. However, many sites presumably associated with early modern humans in these regions are primarily dated using optically stimulated luminescence and uranium-thorium dating methods, which are not included in our database for methodological reasons (see details in Methods). Therefore, the inherent limitation of radiocarbon dating (~ 50,000 years ago[45]) constrained our analysis to outside those regions (Supplementary Fig. 1).

We also observe that the northern route to Mongolia also has a separate pathway to the Altai, Transbaikal, and on to Beringia (~ 34.7 ka; Fig. 1) that led to an entrance into North America starting via the coast of the Pacific Northwest ~ 16.2 ka (Fig. 1, and Supplementary Figs. 4a, 5) and that reached South America no later than 14.8 ± 1.2 ka[46]. This was followed by a second route between the northern sections of the Laurentide and Cordilleran ice sheets[47], through the newly formed ice-free corridor at ~ 13.4 ka, and before the Clovis period[29]. Genetic models of Native American demography agree with our results and reveal a bottleneck during this period, with subsequent population expansion only after 16–13 ka[34]. The peopling of South America followed both (i) a clockwise trajectory crossing Brazil to reach both the east Brazilian and southeast Argentinian coasts, and (ii) an anti-clockwise trajectory crossing the Andes to reach the Peruvian coast and then moving southward to the south of Chile along the Pacific coastline (reaching Patagonia ~14.4 ± 0.2 ka, Fig. 1)[46,48,49]. These results support the hypothesis that humans adapted rapidly to settle at high elevations shortly after their initial entrance into South America[50] via an initial trans-Altiplanic entry from the north[51], or later westward expansions of Amazonian populations into the Andes[52]. However, since our approach mostly captures long-lasting settlement rather than migration pulses (e.g., exit of Africa)[7–10], our results are consistent with expeditions of foragers from the Pacific littoral zone expanding their access to resources[48,53].

Our estimated migration rates (Supplementary Table 2) are higher than those reconstructed for Sahul (0.71–0.92 km year$^{-1}$)[54] or European Neolithic farmers (~ 1 km year$^{-1}$)[55,56], but our results remain plausible. First, our estimates represent the spread of hunter-gatherers that differ from farmers, because the development of farming technology suppresses the expansion rates of more sedentary agriculturalists[57]. Second, human movement depends on the perception of resource availability (or depletion) triggering their next move[58], which often results in either frequent, short-distance movements across warm, highly productive environments, or infrequent, long-distances movements in low-productivity environments. At a continental scale, and assuming humans followed ecosystems of forest-grassland transition (Figs. 1 and 2c), we expect an intermediate strategy of frequent, long-distance movements that would support our results. Third, our estimated expansion rates across western Europe and North America (Supplementary Table 2) match some of those derived from modern-day hunter-gatherer societies that move every three weeks at a highly variable daily travel distance (0.4–15 km year$^{-1}$) depending on the landscape (on/off natural trails)[59,60]. Finally, some of the fastest expansion rates (e.g., across Asia and South America; Supplementary Table 2) could have resulted from some methodological limitations caused by a lack of data in those areas (Supplementary Fig. 1a). An increase in the distribution of available data in those areas would help to refine our local-scale inferences and likely decrease the estimated travel velocity (even though some rare movements of 60–80 km day$^{-1}$ have been recorded[61,62]) (Supplementary Table 2).

## Environmental correlates

We compared the main environmental characteristics of the more- and less-likely routes of expansion by measuring the differences in six environmental variables (Supplementary Fig. 6) between route types at the core of the main hypotheses underlying the peopling of Eurasia and the Americas: (i) mean annual temperature anomalies[21,22,63],

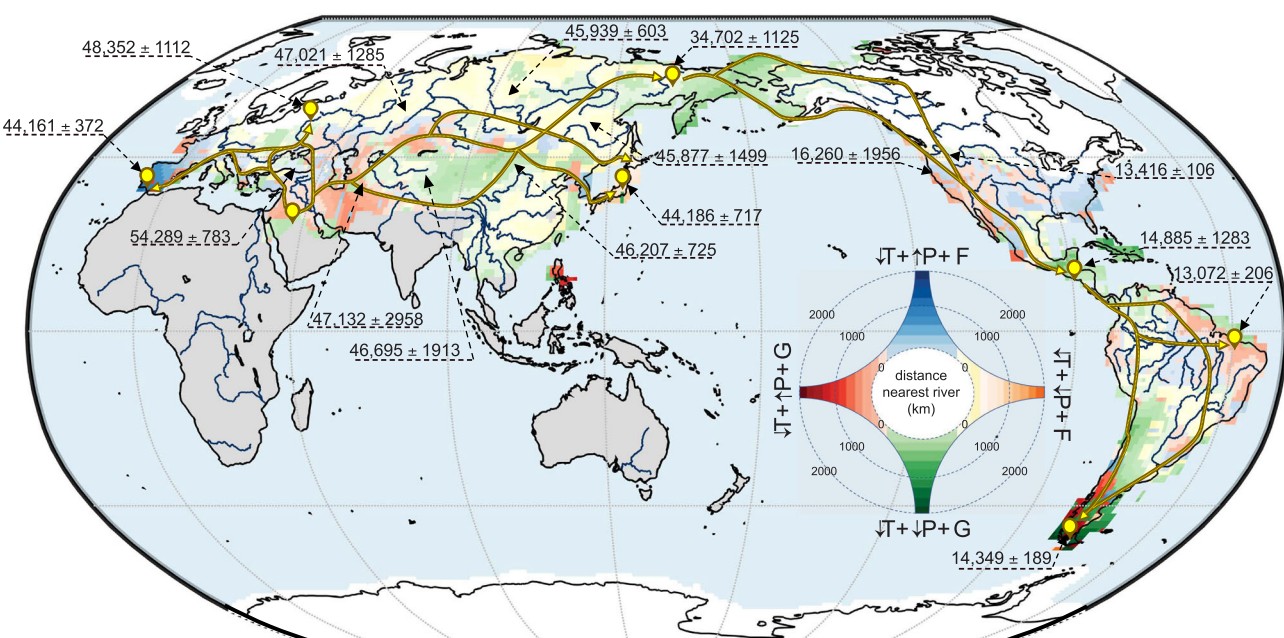

Fig. 1 | **Environmental shifts and human migration pathways.** Regional changes in environmental conditions between the estimated timing of human arrival in a cell (1 ° × 1 ° spatial resolution) and 90 ka (i.e., part of the window when anatomically modern humans permanently exited Africa)[10] categorised by colour gradient as a function of the similarity in pattern of changes relative to 90 ka: red gradient = decrease in mean annual temperature (T) + increase in mean annual precipitation (P) + dominance of grassland (G); blue gradient = decrease in T + increase in P + dominance of forest (F); orange gradient = decrease in T + decrease in P + dominance of F; green gradient = decrease in T + decrease in P + dominance of G. Gradients indicate the distance (in km) of a given grid cell to the nearest river (from near = light red/blue/orange/green to far = dark red/blue/orange/green). Brown-yellow arrows across all panels represent the more-likely routes travelled between sets of location pairs (identified by the brown and yellow icons). White areas indicate the icesheet spread before 16 ka and grey areas show no estimates of timing of arrival because of lack of reliable archaeological data (see details in Methods). Source data are provided as a Source Data file.

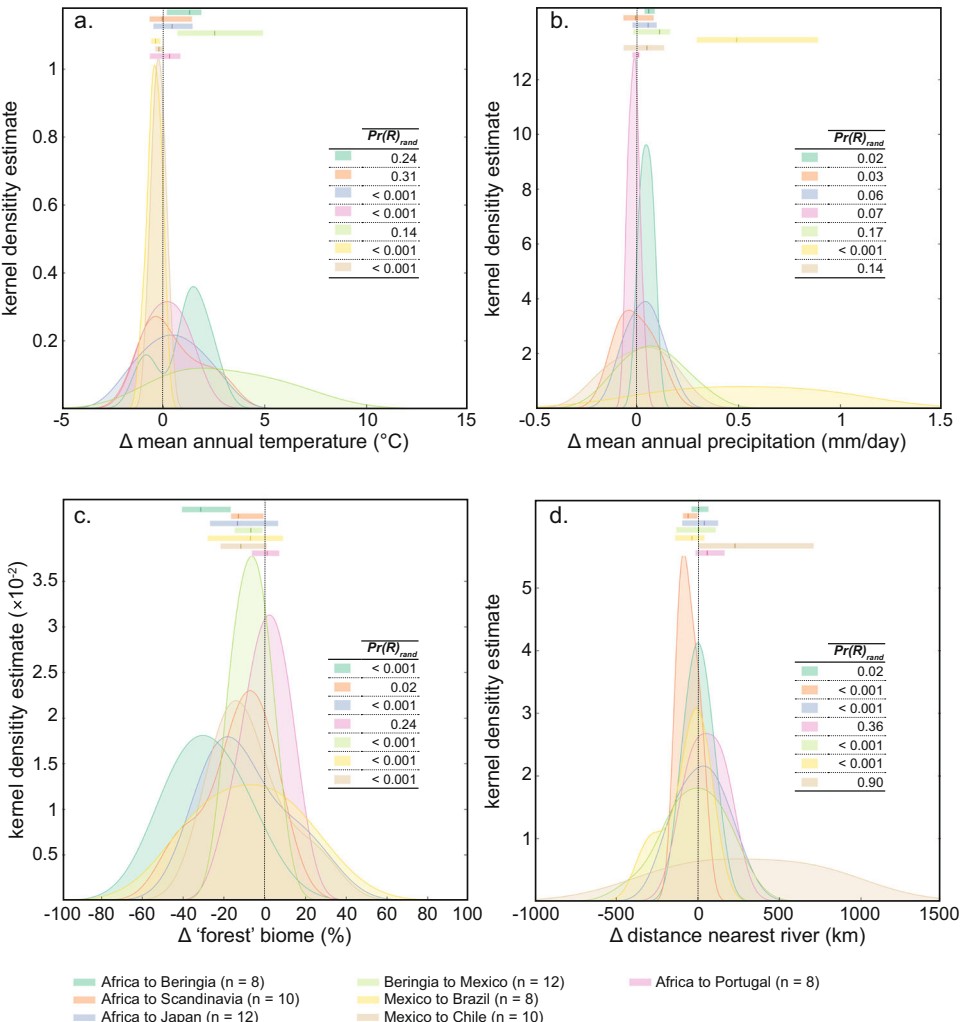

**Fig. 2 | Environmental variability between more- versus less-likely human migration routes.** Distribution (kernel density estimates) of environmental differences (Δ) for each of the *n* set of location pairs (brown-yellow icons in Fig. 1) between the more-likely routes travelled by anatomically modern humans and the other, less-likely routes. For the environmental variables (**a** mean annual temperature anomalies, **b** mean annual precipitation anomalies, **c** percentage of forest, and **d** distance to the nearest rivers), we added on top of each distribution its median (dark marker) and confidence interval (light band calculated as the 25th and 75th percentiles of the distribution) using the respective colour code. The inset table in each panel indicates the estimated probability (Pr(R)$_{rand}$ based on 10,000 random realisations) that regional environmental differences (Δ) between the more-likely routes the other, less-plausible routes is derived by chance. Source data are provided as a Source Data file.

(ii) mean annual precipitation anomalies[2], (iii) proportional vegetation type (forest or grassland biome)[26,64], (iv) landscape ruggedness (landscape accessibility)[23], (v) distance to nearest coast[25], and (vi) distance to nearest river[25]. Based on a randomization test, we then quantified the differences in the mean values of each environmental variable calculated between the more- and less-likely routes to estimate the probability that differences did not arise by chance (see Methods).

Three main environmental variables best explained the differences between the more- and less-likely routes of expansion (probability that the differences did not occur by chance Pr(G)$_{rand}$ > 0.95; Table 1): (i) proportion of forest, (ii) changes in temperature, and (iii) changes in precipitation. These suggest that humans depended on a minimum set of environmental conditions that shaped their expansion across the landscape[2,25,26]. Access to freshwater (inferred here by distance to nearest rivers) also appeared to be an important additional characteristic (Table 1) since many likely expansion routes generally followed major rivers such as the Lena River toward Beringia, both the Dnieper and Danube in Western Europe, the Yellow River in southeastern China, the Mackenzie River and Rio Grande in North America, and both the Amazon and Paraná Rivers in South America (Fig. 1).

However, these environmental variables are intertwined and the dominance of one factor relative to others changes regionally (Fig. 1). Despite the continental decrease in temperature (Supplementary Fig. 7) during the period of human spread across the continents, our results show that humans travelled mainly via the warmer pathways (more likely routes up to 3.5 °C warmer than the other routes with Pr(R)$_{rand}$ < 10$^{-3}$ − i.e, probability that regional climate differences between more- and less likely routes occured by chance − Fig. 2a, Supplementary Fig. 6a) across Eurasia to reach Portugal and Japan. They also followed wetter trajectories (up to 0.5 mm day$^{-1}$ wetter) than the other less-likely routes of migration (Fig. 2b; Pr(R)$_{rand}$ < 0.1), especially to reach Beringia and Japan and across a large part of northeastern South America (Fig. 2b, Supplementary Fig. 6b). The more-likely routes of human migration are also mainly located on (newly) opened landscapes (up to 40% less forest cover; Fig. 2c; Pr(R)$_{rand}$ < 0.1), but the large variance in the estimates of forest loss (up to ± 40%, Fig. 2c) and the geographical locations of these trajectories (Fig. 1, Supplementary Fig. 6d) indicate that humans travelled through grassland areas while staying close to ancient or newly formed forests. These ecosystems of transition, identified as ecotones, likely provided

**Table 1 | Estimated probability (prob = 1 - Pr(G)$_{rand}$) that the difference between environmental values (env variable) along the more-likely routes compared to the other, less-likely routes is not derived by chance**

| env variable | prob | route length (km) | | mean ± sd | | range | |
|---|---|---|---|---|---|---|---|
| | | more-likely | less- likely | more-likely | less-likely | more-likely | less-likely |
| T (° C) | 0.977 | 13,764 ± 7219 | 18,435 ± 7017 | −1.9 ± 0.7 | −2.3 ± 1.4 | 1.9 | 5.7 |
| P (mm day$^{-1}$) | 0.971 | | | −0.01 ± 0.1 | −0.1 ± 0.1 | 0.4 | 0.3 |
| d coast (km) | 0.457 | | | 646 ± 312 | 592 ± 217 | 1005 | 893.8 |
| veg (%) | 1 | | | 37.1 ± 13.4 | 46.1 ± 17.4 | 47.5 | 66.9 |
| d river (km) | 0.902 | | | 278.8 ± 210.6 | 252.6 ± 108.6 | 847.6 | 469.2 |
| rug | 0 | | | 707.4 ± 505.1 | 634.8 ± 719.4 | 1601.8 | 3789.2 |

The probabilities are calculated for mean annual temperature anomaly relative to 90 ka (T), mean annual precipitation relative to 90 ka (P), mean distance to nearest coast (d coast), percentage of forest since 90 ka (veg), mean distance to the nearest main river (d river), and mean ruggedness (rug). We chose 90 ka as an arbitrary baseline because it falls within the timeframe when anatomically modern humans left Africa permanently[10]. We accounted for the effect of the length on Pr(G)$_{rand}$ by normalising the environmental values along each route by its respective length (see details in Methods). Also displayed, the average length of both the more and less-likely routes across all continents along with the mean (± standard deviation) and range (calculated as the difference between the maximum and minimum values) of each environmental variable for all non-normalised trajectories. Source data are provided as a Source Data file.

many advantages to humans because they are areas of relatively high biodiversity[39], have available forests for shelter, wood fuel and food[65], and nearby open habitats facilitate travel and improve visibility for hunting[2,26,64].

Our results did not show major differences in the ruggedness and the distance to the nearest river between the more-likely routes of human expansion and the less-likely routes, except in some parts of South America (Supplementary Fig. 6d). This lack of effect of the ruggedness explains one of the most likely routes crossing the Himalaya and Tibetan Plateau in the middle (Fig. 1) as the fastest way to reach the 4600 m above sea level Nwya Devu in the interior of the Tibetan plateau as early as 40,000 years ago as supported by archaeological evidence[65]. Although further evidence is necessary to support the hypothesis that humans primarily traversed the Himalayas rather than taking alternative paths, these findings indicate that rugged terrain might not have posed insurmountable obstacles to their migration[3,54]. However, the impact of river proximity appeared to be more local in areas where climate conditions became more challenging (i.e., dry or forested habitats; Fig. 1). For example, humans travelled through some of most arid regions in northern Europe by following the path along the Dnieper Rivers (Figs. 1, 2d, with Pr(R)$_{rand}$ < 10$^{-3}$, and Supplementary Fig. 6e). This route aligns with archaeological evidence that identified it as a major conduit of human movement in the early Upper Palaeolithic[66,67]. Meanwhile, the lack of statistical support for the proximity to rivers as a factor for reaching the Atlantic cost of Europe via the Danube (Pr(R)$_{rand}$ = 0.36, Fig. 2d) might be accounted for by the migratory advantage of coastal settings, which were likely leveraged by populations travelling along the Mediterranean Sea (Fig. 1). The pathways connecting the Fertile Crescent with Japan and Beringia via Mongolia presented a similar pattern. Higher rainfall marked the northern path around Mongolia characterised by increased precipitation and either open spaces at the edges of forest or newly formed forests altogether (Fig. 2b, c, Pr(R)$_{rand}$ ≤ 0.02). In contrast, human groups moving through the drier southern routes beneath Mongolia, and subsequently eastward across Russia toward Beringia (Fig. 1, Supplementary Fig. 6a,d) benefited from the proximity to major rivers such as the Lena, Amur, and Yellow Rivers.

Increased precipitation and the dominance of ecotone ecosystems were also essential components for humans to expand into the Americas, first entering along the western coast of North America ~16 ka. However, increasing temperatures from 17 ka[68] led to an opening of the ice-free corridor in the Laurentide ice sheet 3000 years later[69], which created an additional path to North America by 12.6 ka[24] via the Mackenzie River (Figs. 1 and 2a, Pr(R)$_{rand}$ ≤ 10$^{-3}$). These changes in temperature provide the main explanation for the ~ 18,400-years migration hiatus in Beringia before an initial entrance into North America at ~ 16.4 ka (Fig. 1)[17] — almost 10,000 years longer than previously suggested[16,70]. This hiatus is referred to as the 'standstill hypothesis'[71] or the 'Beringian incubation model'[72], and proposes that the ancestors of native Americans remained locally isolated because of ecological barriers such as the large ice sheets that covered North America at the time[63].

The expansion into South America is more complex because of the effort to avoid the generally dry conditions generated by the Antarctic Cold Reversal[73]. The clockwise expansion pattern (Fig. 1) tracked the wettest regions of grasslands bordering the Amazonian forest (Figs. 1, 2b, and Supplementary Fig. 6b,d) along the east coast, as well as exploiting the connectivity provided by the Amazon, Paraguay, and Paraná Rivers to cross more arid or forested regions to reach Chile (Figs. 1, 2d, and Supplementary Fig. 6e). The tributaries of the Amazon River (Fig. 1, Supplementary Fig. 6e) led a westward expansion of Amazonian populations into the Andes despite colder climate conditions (Figs. 1, 2a, and Supplementary Fig. 6b, d) compared to routes from the Pacific littoral or a trans-Altiplanic entry from the north[50].

While subject to the limitations of radiocarbon dating and the spatial distribution of reliably dated archaeological material, as well as assumptions of genetic distance between human populations, our results provide the most objective and robust representation of how and in what conditions humans dispersed so widely around the globe. Including ancient DNA data would provide an independent temporal genetic framework that would strengthen our analysis, but the scarcity of these data at a global scale and for the timeframe considered (i.e., the first human migrations) limits their application in the context of our study.

More importantly, we have provided a robust approach that quantitatively combines genetics, archaeology, and climate modelling into a continuous spatial framework rather than relying on one single type of data or method. Our framework builds on the strength of each discipline while overcoming their inherent limitations to test scenarios that would not be possible to evaluate by relying solely on sparse and incomplete 'snapshot' data of local and spatially isolated events. Moreover, our statistical approach to infer spatio-temporal trajectories of initial human expansion overcomes the methodological limitations of most previous contributions in this area such as: (i) the under-representation of older arrival events[74] and the Signor-Lipps effect[35], (ii) either not spatially explicit[75], or (iii) generating new spatial biases via arbitrary geographic binning[27,76,77], or when interpolating a linear chronology from unevenly spaced age estimates[78], and/or (iv) neglecting uncertainty arising from sampling and taphonomic biases[79], and inherent dating errors[80]. Our results support, at least at broad spatial scales, that the decisions of our early ancestors to penetrate unknown lands would have been driven by suitable environmental conditions. Including Denisovan and/or Neanderthal interactions could potentially be important determinants of the expansion patterns of *Homo sapiens*, but such considerations would require demographic

data for each species (e.g., fertility and survival) and the nature of the interactions among species (e.g., warfare, competition for resources, interbreeding, etc.) that are not available at the spatiotemporal scale we were able to model in this study. This does not preclude cultural decisions from playing a role, but it does demonstrate that spatial scale likely dictates the degree to which environmental determinism can be employed to explain patterns of palaeo-human behaviours.

## Methods

### Archaeological and genetic dataset

We gathered 24,524 radiocarbon-dated archaeological specimens indicating human presence (i.e., human remains, cultural layers or lithic industry, single artefacts, clear hearths, clear butchering, anthropogenic modification, cave art, portable art, living floors, and probable hearths) from four databases (Supplementary Data 1): (i) the Radiocarbon Palaeolithic Europe Database (INQUA: 10,522 ages; ees.-kuleuven.be/geography/projects/14c-palaeolithic), (ii) the Paleoindian Database of the Americas (PIDBA9: 4194 ages; pidba.utk.edu), (iii) the Canadian Archaeological Radiocarbon Database (CARD: 6773 ages; canadianarchaeology.ca), and dates of anatomically modern humans reported in refs. [27] and [18] for Japan and South America, respectively.

The evaluation of the quality of estimated ages is a pre-requisite for any subsequent modelling, inference, and interpretation of records of past life and human impacts[81]. Because of sporadic preservation processes, the oldest dated record in a time series is unlikely to reflect the true timing of an arrival (or, conversely, the youngest dated specimen does not indicate disappearance)[35]. The quality-rating procedure matters because dating techniques, and the laboratory and field protocols supporting them, have been refined over time[82], so the veracity of age/event estimates are subject to continual assessment[83,84]. Given the large number of archaeological specimens to evaluate, we spatially subdivided America and Eurasia into 2.5 ° × 2.5 ° grid cells and focussed on obtaining a quality rating for the five oldest records within each grid cell (those records are the most useful in the calculation of the 'true' timing of initial human arrival accounting for the Signor-Lipps effect) − see details in ref. [80]. We based the quality-rating criteria for these records on ref. [85], with only dates ranked 13 or higher (out of a possible score of 17) deemed acceptable. We used information pertaining to the method of radiocarbon dating (either standard or accelerator mass spectrometry), the type of archaeological evidence associated with the date (e.g., charcoal, whole bone, bone collagen, bone apatite, wood, hide, hair, peat, organic soil, hydroxyproline, shells), the stratigraphic association between the archaeological evidence and the dated material (i.e., direct or indirect association), as well as the type of material dated to assign each record a numbered rank to assess the reliability of the date. We excluded ages if (i) the author(s) stated that the date was anomalous or (ii) had issues with the extraction protocol, (iii) was/were suspicious of contamination of the sample, or (iv) there was not enough information available to assess reliability. When the ages were not directly identified from human remains, but were instead associated with the age of the layer in which artefacts were found (i.e., indirect association), we assumed that any material (i.e., dates and cultural-related technology) coming from Middle Palaeolithic layers (from 300,000 to 50,000 years ago) was associated with Neanderthals, whereas any material (i.e., dates and cultural related technology) from the Upper Palaeolithic (from 50,000 to 12,000 years ago) was associated with anatomically modern humans[86]. We disregarded any material in the transition from Middle to Upper Palaeolithic since there are transitional industries whose species attribution is controversial. Grounds for excluding dates from controversial South American sites are detailed in ref. [18]. This filtering resulted in a total of 5977 reliable specimen ages associated with human presence (Supplementary Data 1). We calibrated all $^{14}$C dates to calendar years before present using the Northern (IntCal13) and Southern (ShCal13) Hemisphere Calibration curves[84,87] from the OxCal

radiocarbon calibration tool Version 4.2 (see https://c14.arch.ox.ac.uk).

We gathered a total of 67,643 human mitochondrial control region sequences from Genebank (last accessed 28 May 2017, the last available update in the regions of interest). We excluded ancient DNA data because they are too scarce or even non-existant in most regions of the world to run short tandem-repeat analyses and obtain enough nuclear coverage to address these movements of early modern humans. We then retained 27,506 sequences with reliable geographical locations to compute the fixation index ($F_{ST}$), a proven and robust method to infer genetic distance between populations[88]. Mitochondrial haplogroups were classified from control region sequences using HaploGrep 2[89], according to nomenclature provided by PhyloTree mtDNA tree Build 17 (phylotree.org; ref. [90]), giving a total of 31 haplogroups (A, B, C, D, E, F, G, H, I, J, K, L0, L1, L2, L3, L4, L5, M, N, O, P, Q, R, S, T, U, V, W, X, Y, Z). We sorted the data by geographical location by grouping the samples according to 96 geographical locations, which were either country or region/province. We then converted the data into Genpop format using the R package `pegas` and generated the matrix of pairwise $F_{ST}$ between the 96 geographical locations using haplogroup frequencies[91] and the R package `diveRsity`. Ultimately, we removed non-Indigenous haplogroups.

### Inferring the regional timing of initial human arrival

We used a maximum-likelihood method first developed in ref. [92] to correct for the Signor-Lipps effect that we adapted for spatial inference of human settlement patterns. The original version of the model assumes that the true ages (e.g., estimated using radiocarbon techniques) of specimens within a time series, $T_1,...,T_n$, are independent and uniformly distributed over the interval ($T_{ext}$, $\gamma$), which correspond to extinction ($T_{ext} = 0$, i.e., the present day because *Homo sapiens* is not extinct) and settlement times ($\gamma$) for the species under investigation, respectively. In contrast to the original version of this model that assumes a constant dating error across samples, we assumed that the radiometric errors associated with each estimated age were approximately Normally distributed[82]. Thus, $\hat{T}_k$ estimates the $k^{th}$ age assuming the true age $T_k$ follows:

$$\hat{T}_k \sim g\left(T_k, \sigma_{(k)}^2\right) \tag{1}$$

with $g$ being the Gaussian probability density function describing the radiometric error $\sigma$. The estimated timing of human colonisation $\hat{\gamma}$ is then calculated by numerically maximising the log-likelihood $\mathcal{L}(\gamma)$ over $\gamma$:

$$\mathcal{L}(\gamma) = \sum_{k=1}^{n} \log h\left(\hat{T}_k\right) \tag{2}$$

with $h$ being the probability density of $\hat{T}_n$

$$h\left(\hat{T}_k\right) = \int_{\gamma}^{T_{ext}} g(t, \sigma_k^2) \frac{dt}{T_{ext} - \gamma} \tag{3}$$

To adapt this approach to questions of spatial inference, we defined $W$ as the spatial landscape, and $\gamma_{(x)}$ as the date of first human arrival at a given location $x$ in $W$. We assumed that $\hat{T}_1,..., \hat{T}_n$ are individual point estimates of human presence based on independently discovered, dated specimens found at location sites ($x_1,..., x_n$). According to Eq. 1, the estimated timing of human arrival $\hat{\gamma}(x)$ at a given location $x$ is calculated by numerically maximising the weighted likelihood $\mathcal{L}_x(\gamma)$[93] over $\gamma$ across $W$:

$$\mathcal{L}_x(\gamma) = \sum_{k=1}^{n} w(x - x_k) \log h_{\gamma}\left(\hat{T}_k\right) \tag{4}$$

where $w(x - x_k)$ is a weighting factor, such that $\sum w(x - x_k) = 1$. The standard procedure in estimating local density is to select a weighting factor proportional to:

$$w(z) = \frac{1}{b} g_0 \left( \frac{z}{b} \right) \tag{5}$$

where $g_0$ = the density of the standardised Gaussian distribution and $b$ is an optimised bandwidth to prevent any spatial bias in $\hat{\gamma}(x)$ because the dated specimens are not uniformly distributed across space (Supplementary Fig. 1a). This spatial bias therefore depends on the size of the bandwidth $b$, because if $b$ is too large, included specimens no longer follow the statistical law at the given location $x$; however, if $b$ is too small, local variance becomes too high because of a shortage of samples.

The main idea is to find a trade-off in the size of $b$ to obtain both the lowest local bias and local variance[94,95]. We corrected for this bias by using a simulation-based spatial bias-correction procedure (Supplementary Fig. 2). This procedure estimates the bias generated by the model at each grid cell (along with its variance across space) given an optimised bandwidth size. The first step (model inference) calculates a preliminary spatial estimate from the model for each grid cell for a predefined bandwidth size (arbitrarily set to one tenth the maximum pairwise distance between the data at each location). In the following steps, we assumed this 'preliminary' estimate to be a true date of occurrence $\gamma_{(p)}$ for each grid cell. Based on these $\gamma_{(p)}$, we generated $n$ simulated time series ($n = 100$, step 2) following the same spatial location and characteristics (i.e., number of ages, laboratory dating error; see details in ref. 80 as for the dated record). We then inferred for each of the $n$ simulated datasets a spatial estimate of timing of occurrence per grid cell using the model (step 3), and we calculated the average estimate for each grid cell across the $n$ simulated time series (step 4). By comparing the average estimate to $\gamma_{(p)}$, we calculated a mean bias $B_{\hat{\gamma}_{(x)}}$ in each grid cell $x$ and its associated variance $\sigma^2_{\hat{\gamma}_{(x)}}$ across space that are used to approximate the integrated mean-squared error across the landscape $W(E)$:

$$E \simeq \sum_x \left( \sigma^2_{\hat{\gamma}_{(x)}} + B^2_{\hat{\gamma}_{(x)}} \right) \tag{6}$$

We then repeated steps 1–4 using different bandwidth sizes until we obtained the lowest $E$ (steps 5 and 6). In the final step, we subtracted this bias spatially to $\gamma_{(p)}$ in each grid cell to provide a final and spatially corrected estimate of occurrence timing (step 7) given the optimal bandwidth. By integrating both the temporal and spatial distributions of the radiocarbon dates and including their standard dating error into our probabilistic framework, we (i) correct for the Signor-Lipps effect in a spatially explicit way, and (ii) return reliable estimates of the timing of arrival at times even older than the limit of reliable, direct inference possible from radiocarbon methods.

We evaluated the ability of our statistical approach to reproduce the regional estimates of simulated (i.e., set by us) timings of first occurrence. First, we generated simulated prediction surfaces of timings of initial occurrence following two main scenarios: (1) the first scenario describes a gradual peopling across space (Supplementary Fig. 3, lower-left panel) whereas the second scenario describes two entrances to the landscape (Supplementary Fig. 3, bottom-right panel). We called these surfaces 'benchmark maps'. Next, for each timing of occurrence we simulated time series of ages assuming a uniform distribution, along with their associated standard deviations, such that older ages have a larger standard deviation. These time series represent dated archaeological specimens. Thus, we applied our statistical approach to these simulated archaeological specimens to infer a 'new map' of regional timing of initial arrival. We compared this new map to

the original benchmark map. We repeated the entire process one hundred times to account for potential variability in our results.

## Human-expansion trajectories

We calculated human movement pathways between seven sets of two locations each (i.e., a source and a destination that represented the extremes of movement pathways across a continent, see brown-yellow icons in Fig. 1 and Supplementary Fig. 5) across continents gridded at a spatial resolution of 1° × 1°. These locations were based on two independent proxies: (1) the estimated regional timing of first human arrival from archaeological evidence, and (2) mitochondrial-based $F_{ST}$ pairwise distances (see details of these proxies in Section 1). The seven pairs of 'source-destination' locations (Supplementary Fig. 5) are: (i) Fertile Crescent and Beringia, (ii) Fertile Crescent and Scandinavia, (iii) Fertile Crescent and Japan, (iv) Fertile Crescent and Portugal, (v) Beringia and Central America, (vi) Central and Eastern Brazil, (vii) Central America and Chile. For each pair of source-destination locations, we evaluated multiple trajectories quantified as a function of a cost ($pT$) according to the following equation:

$$pT = \sum_{i=1}^{n} e^{-\alpha \Delta^2} \frac{e^{\beta \Delta}}{1 + e^{\beta \Delta}} \tag{7}$$

This cost is the sum of individual cost $e^{-\alpha \Delta^2} \frac{e^{\beta \Delta}}{1 + e^{\beta \Delta}}$ to move from one grid cell to the next along the $n$ grid cells of a trajectory between the source and the destination. For a given trajectory and depending on the proxy we used (i.e., archaeological or genetic evidence), $\Delta$ is either the difference of timings of first human arrival, or the pairwise distance value of mitochondrial $F_{ST}$ between a grid cell and its neighbour. Since the general algorithm aims at selecting neighbouring grid cells with small differences (i.e., $\Delta$) either in the timing of human occurrence or in genetic distances ($F_{ST}$), $e^{-\alpha \Delta^2}$ ensures that neighbouring grid cells with high $\Delta$ are not selected, whereas $e^{\beta \Delta}$ promotes selecting neighbouring grid cells with low $\Delta$. However, least-cost paths have been criticized because of the implicit assumption of a goal-oriented search between locations, i.e., that humans would take a single, optimal, least-cost path. This assumption would disregard any random exploratory behaviours and the likelihood that long-term, most-travelled corridors might not be the result of a single, least-cost path, but instead multiple pathways of lesser cost ('least-cost' corridors)[96]. By authorizing the selection of lesser goal-oriented pathways, some emergent 'optimal' trajectories might not always be the 'least cost' options.

The evaluated trajectories (Supplementary Fig. 5) included the 'optimal' trajectory between a source and its related destination (in blue and orange; Supplementary Fig. 5) and other plausible scenarios given the space to travel (in green; Supplementary Fig. 5). The optimal trajectory aims to explore the potential movement space using a Metropolis simulated annealing algorithm[97,98] to return the path between the source and its destination with the lowest $pT$. The spatial exploration algorithm proceeds as follows: (i) we determined an initial path that is the shortest distance between the source and its related destination; (ii) we calculated the cost $pT$ of this initial path (we call this $pT_1$); (iii) we randomly selected one grid cell along this path and modified its location to be one of its six available neighbours (i.e., excluding the former and the next grid cell in the path); (iv) we calculated the $pT$ of this new path (i.e., $pT_2$) — if $pT_2 < pT_1$, we adopted the new path and the algorithm started again at step iii; otherwise, the rejection of the new path is conditional on $A$, an accept/reject probability calculated as:

$$A = e^{\left( \frac{-(pT_2 - pT_1)}{kT} \right)} \tag{8}$$

If $pT_2 > pT_1$, we generated a uniform random number $u$ between 0 and 1 that we compared to $A$. If $u < A$, we accepted the new path; otherwise, we rejected the new path, and we defaulted to the former

path and modified the location of another grid cell to recalculate a cost (i.e., returning to step iii). $A$ decreases as new patterns are accepted and increases as new patterns are rejected. This space-searching process stopped when a set number of successive iterations $kT$ (with a maximum of 10,000 chosen arbitrarily) failed to decrease $pT$. This algorithm presents two main advantages compared to a classic least-cost path approach because it first introduces some randomness to mimic human behaviour (i.e., assuming that human decisions are not only based on cost-efficiency reasoning), and it guarantees a global spatial minimisation of $pT$ (as opposed to a local minimisation of $pT$).

For each pair of source-destination locations, we ranked the trajectories to select one main trajectory. Ranking was a function of $pT$ and the agreement between the two types of proxy used to calculate them − the main trajectory was the one with the lowest $pT$ based on archaeological data that also had the lowest $pT$ for mitochondrial $F_{ST}$ pairwise distance. If another trajectory presented a $pT$ of a similar order of magnitude as the one with the lowest $pT$, we also selected that trajectory.

### Environmental variables and randomisation tests

We established a hypothetical framework potentially explaining variation in initial movement patterns based on six environmental variables. These were: (i) distance to nearest major river (see the list of these rivers below), (ii) distance to the nearest coast, (iii) mean annual temperature anomaly (at the estimated timing of arrival; Supplementary Fig. 4a) relative to 90 ka, (iv) mean annual precipitation anomaly (at the estimated timing of arrival; Supplementary Fig. 4a) relative to 90 ka, (v) present-day ruggedness of the landscape, and (vi) the percentage of forest *versus* grassland (at the estimated timing of arrival; Supplementary Fig. 4a) along the trajectory.

We hypothesised that the main trajectories would be closer to rivers than other plausible routes because of the necessity of access to predictable and perennial water sources[25]. These major rivers include: Danube, Volga, Obs, Lena, Yenisei, Yangtzee, Mekong, Yellow, Amur, Mackenzie, Columbia, Amazon (and its main branches), Paraguay, and Paraná. We also hypothesized that coastal movements (i.e., pathways with the shortest distance to the nearest ocean/sea) would be favoured over paths more distant from the ocean given the availability of additional seafood resources and the safety from habitat disruptions faced by inland populations during the rapid climatic fluctuations of the Late Pleistocene[99,100]. We calculated the distance of each grid cell to its nearest coastline over the last 120,000 years at a 1000-year interval. We accounted for the changes in sea level over time by iteratively applying a change of land-sea mask derived from a global sea-level record[101] overlaid onto present-day coastlines taken from the ETOPO1 dataset[102]. For the temperature and precipitation anomalies, we hypothesized that the routes would be more likely to occur in regions with mid-range temperature and precipitation conditions[2,25]. We used the already published mean annual temperature and precipitation simulations for the last 120,000 years from the HadCM3 Atmosphere-Ocean General Circulation Model[103]. The HadCM3 climate model also provides monthly and seasonal temperatures and precipitation, but we excluded these variables from the analysis to avoid multicollinearity effects[104]. To capture the broad-scale climate dynamic at the time of human expansion[105], we calculated the values of mean annual temperature, and mean annual precipitation at the estimated timing of arrival in each grid cell (Supplementary Fig. 4a) and relative to 90 ka (see Supplementary Fig. 6a, b respectively). We hypothesised that ruggedness would limit human movements, such that the main pathway should follow regions of lower ruggedness than elsewhere (i.e., indicating ease of travel). A ruggedness index is used to analyse the topographic heterogeneity of a terrain, and we computed this index[106] as the difference in elevation between a given cell and its 8 neighbouring central cells (Moore's neighbourhood), based on a digital elevation model (ETOPO1 global relief model of the Earth's surface)[102].

Then for a given cell, we squared each of the eight elevation differences (to make them positive) and calculated the square root of the average square. Finally, we hypothesised that mid-range forest/grassland ecotones would provide the ideal combination of shelter (forest) and hunting opportunities (grassland), because ecotone habitats tend to harbour higher diversity and biomass than non-ecotone habitats[39]. We used the global simulations of vegetation changes over the last 120,000 years from the BIOME4 vegetation model forced offline using the HadCM3 climate simulations[103,107]. The BIOME4 model returned for each grid cell the dominant functional type among 28 biome types that we then reclassified into two categories: forest or grassland (see classification correspondence in Supplementary Table 1). BIOME4 projections at 0, 6 and 18 ka have already been successfully evaluated at the biome level against mapped palaeo-vegetation patterns at the same times using fossil pollen and plant-macrofossil data, extracted from the BIOME6000 database Version 4.2 (bridge.bris.ac.uk/resources/Databases/BIOMES_data)[107]. We re-evaluated BIOME4 projections for the same time periods (0, 6 and 18 ka) for the vegetation types (defined as forest or grassland) using both BIOME4 outputs and BIOME6000 reconstructed vegetation (Supplementary Fig. 8). We also used Supplementary Table 1 to classify BIOME6000 data into vegetation types (i.e., forest or grassland). Ultimately, we calculated the changes in simulated dominant vegetation type (forest or grassland biome) at the estimated timing of arrival in each grid cell (Supplementary Fig. 4a) and relative to 90 ka (see Supplementary Fig. 6d).

We retained the value of each environmental variable at the estimated inferred date of initial arrival for each grid cell along each trajectory: mean annual temperature, mean annual precipitation, distance to the nearest coast, and type of biome (Supplementary Fig. 4a). To evaluate whether a given environmental variable is a potential driver of the primary human-expansion pathways (blue and orange trajectories in Supplementary Fig. 5) compared to other trajectories (green trajectories in Supplementary Fig. 5), we constructed a global and a regional randomisation test (Table 1 and Fig. 2, respectively). The global randomisation test (Table 1) compares all trajectories across all regions while taking into account their relative area. For each environmental variable (Supplementary Fig. 6), we calculated the average value along each trajectory multiplied by the total length of the trajectory to represent the total effect of a given variable on each trajectory. These averaged environmental values are normalised by region (because each region is a different size) to make all trajectories across all regions comparable. For each environmental variable, we (i) calculated the difference ($d_1$) between the averaged values of non-selected trajectories (green trajectories in Supplementary Fig. 5) minus the averaged values of the main trajectories (blue and orange trajectories in Supplementary Fig. 5), (ii) randomly reattributed the type of trajectories (i.e., main trajectory *versus* other trajectories) across all regions and recalculated the difference ($d_{rand}$), (iii) repeated this process 10,000 times to obtain 10,000 $d_{rand}$, (iv) summarised the randomisation results by calculating the probability ($Pr(G)_{rand}$ = global probability of returning the result by random after correcting for each region's area) that the proportion of all $d_{rand} > d_1$, counting the occurences of $d_1$ in the denominator and the numerator. The higher the $Pr(G)_{rand}$, the more likely it is that the difference in a given environmental variable between the main trajectories and the other trajectories occurs by chance. The regional randomization test measures the environmental differences across all trajectories within a specific region. Adhering to the methodology of the global randomization test, we computed the probability ($Pr(R)_{rand}$) for each region and environmental variable (Fig. 2, Supplementary Fig. 6). Specifically, we determined the likelihood (i.e., higher $Pr(G)_{rand}$ = more likely) that the median pairwise difference in any environmental variable between the main trajectories and alternate trajectories could arise by chance.

## Main assumptions

We acknowledge that by focusing on the northern route of expansion in Eurasia via the Fertile Crescent, we cannot address the major debate about the relative roles of coastal expansion and environmental adaptation along the southern spread route in Eurasia (via Arabia and south Asia). We are not discarding the role of a southern route in the expansion of *Homo sapiens*, but considering this route would generate two methodological problems.

First, many sites presumably associated with early modern humans in the southern Asia regions are primarily dated using optically stimulated luminescence and uranium-thorium dating. Given the differences in the dating process and the nature of the age uncertainties of these approaches (i.e., magnitude, source of variance, etc.)[108], integrating multiple types of data into a common modelling framework would be challenging and most likely decrease the robustness of our estimated timing of arrival. While we acknowledge that our method should be further expanded with additional data from all chronometric methods to investigate the southern Eurasia route, we argue that in the context of the northern expansion route of *Homo sapiens*, the small proportion of these additional data relative to the 5977 reliable radiocarbon dates we used here is unlikely to alter our results.

Second, these ages estimated using optically stimulated luminescence and uranium-thorium methods often describe the presence of anatomically modern humans older than 50,000 years ago, which challenge our assumption that the Middle Palaeolithic is associated with Neanderthals, whereas any material from the Upper Palaeolithic was associated with anatomically modern humans[86]. We concede that before 50 ka, *Homo sapiens* could be associated with Middle Palaeolithic industries[109,110], but the fact that our dataset is only based on radiocarbon dates means that we are confident that the Middle Palaeolithic is associated with Neanderthals[111], except in a few regional cases (e.g., Chatelperronian of northern Spain/France).

Ultimately, focusing on the northern route of expansion resulted in excluding the following countries in our study that are primarily dated using optically stimulated luminescence and uranium-thorium methods: Saudi Arabia (2,149,690 km²), Yemen (527,968 km²), Oman (309,500 km²), United Arab Emirates (83,600 km²), Pakistan (881,912 km²), India (3,287,590 km²), Nepal (147,181 km²), Bangladesh (147,570 km²), Myanmar (676,578 km²), Laos (236,800 km²), Thailand (513,212 km²), Vietnam (331,212 km²), Cambodia (181,035 km²), Malaysia (330,803 km²), Indonesia (1,904,569 km²), and Philippines (342,353 km²). The sum of these excluded land area (~ 8,980,723 km² represents) ~9.5% of our entire study area (i.e., Eurasia + Americas = ~ 94,000,000 km²). We therefore argue that despite the importance of the south Eurasian route of expansion, our results still show the role of environmental adaptations (including coastal changes such as in the Pacific Northwest and northern Europe) to facilitate/hinder the spread of modern humans across > 90% of both Eurasia and the Americas.

## Reporting summary

Further information on research design is available in the Nature Portfolio Reporting Summary linked to this article.

## Data availability

Source data are provided with this paper and all data are also available for download at github.com/FredSaltre/HumanGlobalExpansion and at https://doi.org/10.5281/zenodo.11078937.

## Code availability

All R and Matlab code (including source data files) are freely available at github.com/FredSaltre/HumanGlobalExpansion and at https://doi.org/10.5281/zenodo.11078937.

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

## Acknowledgements

We thank B. Hoogakker (University of Oxford), J. Singarayer, and R. Smith (University of Reading) for providing the simulated BIOME4 data and the palaeoclimate reconstructions from the HadCM3-forced BIOME4 simulations. This research was supported by the Australian Research Council (ARC) Centre of Excellence for Australian Biodiversity and Heritage (CE170100015) to C.J.A.B. B.L. was supported by an ARC Future Fellowship (FT170100448). We acknowledge the Indigenous Traditional Owners of the land on which Flinders University is built—the Kaurna people of the Adelaide Plains.

## Author contributions

All authors contributed to drafting the manuscript. F.S. designed the study, developed the model, analysed the data and led the writing. J.C. helped develop the models and statistical analyses. S.B. and E.B. compiled, vetted, and cleaned the archaeological dataset. T.H and M.O. helped analyse the human expansion based on archaeological evidence. B.L. provided the genetic dataset and helped interpret the human expansion trajectories based on genetic evidence. C.J.A.B. helped design the study and interpret the results. All authors gave final approval for publication.

## Competing interests

The authors declare no competing interests.
