## [Peer Review File · Nature Communications]

Environmental conditions associated with initial northern expansion of anatomically modern humansReviewers' Comments:

Reviewer #1:

Remarks to the Author:

Understanding the peopling of various continents and regions of our species is for sure an interesting and also a key topic in the studies of human evolution. Archaeological evidence is accumulative which means the earliest evidence of modern human presence in a certain region updates quickly through time. Modeling then becomes a complementary and sometimes efficient way to understand the dispersal of modern humans. In the manuscript, Dr. Saltré et al compiled a huge number of radiocarbon dates and genetic data, and developed a statistical approach to indicate the most-likely initial expansion routes of modern humans in Eurasia and the Americas. The results provide us a new scenario understanding the dispersal of early modern humans in the northern part of the old world and the new world. However, we should also be aware of that those routes are more likely presumptive for that it works only if the archaeological and genetic data are both complete and accurate which may not be always the case. And this shortage of archaeological data has been noticed by the authors.

I am sure there are other reviewers who will check the mathematics and statistics of the simulation. Here I will comment on the archaeology mainly.

1. The title of the manuscript may not be appropriate for that the authors mainly discussed the northern route of the dispersal, while the southern routes which are more familiar by many readers are not included.
2. The following comment relates to the first one. Without the data of the southern route, the reasons shaped the patterns of initial expansions of modern humans probably only limit to the discussions of the northern part of the Old World.
3. When the authors said there are not enough data in the southern part of the Old world, it may due to the dating method the authors have chosen. Many of the sites presumably associated with early modern humans in those regions were dated to MIS 5 and 3 mainly by OSL and U-series methods. Some of the sites have been mentioned by the authors in lines 52-54.
4. About the main results of this paper, there are a lot of room to discuss. The authors conclude that cultural drivers remain plausible at a finer scale, the migration corridors are predominantly constrained by regional environmental conditions. "Migration corridors are constrained by environmental conditions" seems like a common sense. It might be more important to learn why people moved at certain time from a region to another, and the various constrains among different regions. For instance, people arrived Far East at around 35 ka, but they moved forward largely after LGM. The common explanation is that the Bering land bridge hypothesis. However, LGM or other harsh environments may hinder the expansion of hominin groups, such as the refugia hypothesis in Europe. Migrations of modern humans from Levant to Europe are complicated as well when taking the earlier dispersals into account.
5. Lines 259-262. In the method part, I admire the efforts putting by the authors to sort out the reliable radiocarbon dates. However, the assumption that Middle Paleolithic (300-50 ka) was associated with Neanderthals and Upper Paleolithic (50-12 ka) was associated with modern humans was not clear. Did the author use dates as means to sort different hominin groups out, or both dates and cultures (Middle Paleolithic or Upper Paleolithic technology)? If only dates, it is not reasonable. If both, it is better to make it clear. Denisovans in the Altai Mountain area and the edge of the Tibetan Plateau make things even more complicated.
6. Line 271. Although I am not an expert studying genetics, the reason to exclude ancient DNA data is not easy to understand, at least based upon the current manuscript. And I am wondering whether the result will be different or not if including ancient DNA as we know the results indicated by the ancient DNA and by mitochondrial haplogroup are not always the same.
7. Lines 378-379. Coastal movements would be the ones distant from the ocean? Distant or adjacent? When talking about the coastal movements, sea level fluctuations should be considered.
8. The routes in Fig. 1 may need some explanations. For instance, Fig. 1 shows that modern humans crossed the Himalaya and Tibetan Plateau. It seems not a least cost way in many readers.
9. A few minor comments. F column of the excel table is not countries.

Reviewer #2:

Remarks to the Author:

Environmental conditions shaped the patterns of initial expansion by anatomically modern humans

Frédéric Saltré, Joël Chadœuf, Thomas Higham, Monty Ochocki, Sebastián Block, Ellyse Bunney, Bastien Llamas, Corey J. A. Bradshaw

=====

I could only read the manuscript briefly and not go in detail into the supplementary material (this was a caveat I highlighted when accepting the request to comment on this article), so please take my comments with a grain of salt.

I find the study interesting. From a brief skim it seems to me that the methods are well-explained and that some serious work has been put into this study. The results make a lot of sense, and it is nice to see a methodological, computational method applied to the assessment of this question, based on a large dataset that brings together genetic and carbon-dating approaches.

The genetic analysis seems pretty simplistic, but this may be an advantage at such a broad scale and with a lot of data.

I'm not sure how novel the findings are and whether the study really succeeds to achieve its goals. Some comments that perhaps the authors might be able to answer and could help in clarifying these questions:

1. The archaeological reconstruction is based on radiocarbon data? I'm no expert, but I think the limit of reliable inference of radiocarbon is ~40kya, and so founding on it the reconstruction of events that mostly happened earlier is odd (in other words: this should seemingly be reliable only in the Americas, and not in the spread most of the rest of the world, which happened earlier).

2. The dichotomy presented in the paper's narrative seems a bit like a false dichotomy: I don't think anyone suggests that environment didn't have a major influence on the spread of humans across the world. I find it hard to imagine a scenario in which the routes of human spread are not least-resistance routes across the landscape with respect to some environmental variable, regardless of whether cultural or other adaptation to local environments did or didn't occur in the process (or whether any other cultural dynamics, e.g. inter-group competition played a role, or cultural selection on dispersal, or within-group competition for resources, or anything like that. Even if such factors were the main drivers of expansion, it would still have been expected to occur along routes that are more convenient in their environmental variables). The debate / open question might be (and is, I think) which environmental variables mattered, and which didn't (or more precisely: which more and which less).

3. If one really wants to tease apart the roles of cultural / population dynamics from the role of environmental constraints on spread, perhaps the way to go would have been (and is clearly not within the scope of the current study) checking whether the specific timing of bursts of spread is aligned with climatic fluctuations that would have opened/closed certain environmental corridors (such studies have / are being done; I'm not aware of the details, though). This is associated with many challenges, of course, and will also have its limitations.

4. Continuing from (2), one would expect the study to explore a large number of environmental variables and highlight those that were more/less important; but (again, my reading was not as thorough as I'd like, I apologize if I'm wrong here) it seems that the conclusion is that pretty much

all/most of the variables that were explored turned out to be important (perhaps because they are inevitably non-independent of one another).

5. Furthermore, there's been a major debate about the role of coastal expansion and a southern route of spread (Arabia, south Asia), but the data doesn't allow to test this, so a major aspect of the debate about the role of environmental adaptation to facilitating/hindering spread could not be addressed.

In sum – I like this study, but I'm not convinced that it is a breakthrough in our understanding of human spread, and the novelty isn't completely clear to me (beyond the fact that it was done in what seems to be a methodical approach with orderly and systematic treatment of a large dataset, which are important).

Perhaps a different pitch or some clarifications could settle these open questions, but maybe not – I'm sincerely not sure. I'm sure the study will be published, and I'll be happy to see it in the literature; I'm not sure whether Nat Comms is the appropriate venue or not. (I mean this literally; I'm not saying it isn't, I'm saying I can't tell whether it is or is not).

=====

Line 169: The "incompatible" seems like it isn't in place.

Reviewer #3:

None

Reviewer #4:

Remarks to the Author:

Saltre et al used archaeological and genetic data to predict the most likely path taken by modern humans to be spread in Eurasia and America. The path they predicted broadly agrees with what is already known by genetic data, but combining with archaeological data, they can make more robust inferences. Furthermore, by looking at the environmental variables of these paths, they can predict what might be the essential factor for human dispersal. Although the methods used here are fascinating, I have some points before it is acceptable for this journal.

Major:

1. The authors said they have only used modern mitochondrial data for this analysis. However, as we know that there are several substantial population replacements of indigenous populations (at least in Europe and Middle East) I am not sure that using only modern data is enough for mitochondria. Especially as the authors deliberately removed all the ancient genome, which might be a better representative of the actual ancient mitochondrial status of the region rather than the modern population. Authors should either use ancient genomes or discuss why ancient genomes were removed in this case.

2. Although Figure 1 is generally quite good in representing the human out of Africa migration with appropriate time points, I have noticed some results are a little bit debatable. Especially a human population was migrating from Central Asia to Japan (close to 10,000 km) in less than 1,500 years. The authors should discuss some possible drawbacks of such a fast expansion rate (for example, underestimation of time in Central Asia due to unavailability of archaeological data). A supplementary Table will be appreciated with the distance covered and time taken for different paths (Africa to Beringia, Beringia to Mexico etc.). Also, some citations about the expansion rate and if the authors' results match that.

3. In Page 8 line 206: "such as in the driest part of Scandinavia and Western Europe via both the Danube and the Dnieper Rivers": I am not sure why the mean and confidence interval of Scandinavia and Western Europe shows two different directions for the nearest river. Whereas Western Europe shows a positive tendency, Scandinavia shows a negative tendency. Can the author explain it better

for the opposite results?

Minor:

1. The F_{st} calculation is not clear for mitochondrial data. Are they directly calculated by the distance between the mitochondrial genome? Or are they calculated from the known distance coming from haplogroup information?
2. Similarly, in case there are several individuals from the same regions, how the F_{st} is calculated for them?
3. Extended Figure 1b: the dots, if they represent mitochondrial DNA samples, do not look like there are 27 thousand samples. Please clarify. Also, a better description is needed for the caption.
4. Is there any particular reason to use such an older dataset for Genbank (28 may 2017)?
5. It would be better to have a little bit more time data point in Figure 1. For example, the separation time between European and East Asian in the Middle East (the yellow dot).
6. Page 8 line 201: Not sure what is meant by Fig. 3c. I can see only two main figures.
7. Figure 2: It would be great if the authors could sort the median and confidence interval accordingly, as it is mentioned in the footnote. Easy to find which colour corresponds to which path.
8. Page 8 line 207: (i.e., dry or forested habitats; Fig. 2a). I thought Figure 2a represented temperature, not dryness.

Environmental conditions shaped the patterns of initial expansion by anatomically modern humans

Reviewer #1

Understanding the peopling of various continents and regions of our species is for sure an interesting and also a key topic in the studies of human evolution. Archaeological evidence is accumulative which means the earliest evidence of modern human presence in a certain region updates quickly through time. Modeling then becomes a complementary and sometimes efficient way to understand the dispersal of modern humans. In the manuscript, Dr. Saltré et al compiled a huge number of radiocarbon dates and genetic data, and developed a statistical approach to indicate the most-likely initial expansion routes of modern humans in Eurasia and the Americas. The results provide us a new scenario understanding the dispersal of early modern humans in the northern part of the old world and the new world. However, we should also be aware of that those routes are more likely presumptive for that it works only if the archaeological and genetic data are both complete and accurate which may not be always the case. And this shortage of archaeological data has been noticed by the authors.

I am sure there are other reviewers who will check the mathematics and statistics of the simulation. Here I will comment on the archaeology mainly.

1. The title of the manuscript may not be appropriate for that the authors mainly discussed the northern route of the dispersal, while the southern routes which are more familiar by many readers are not included.

Response 1: This is a valid point. We tried to make the title as general as possible because although it is correct that we excluded the southern routes, those routes only represent < 10% of the total land mass of our study area. We have now addressed this point in detail in the *Supplementary Methods*, *Supplementary Discussion*, and the main text (see **Response 14** for more details).

2. The following comment relates to the first one. Without the data of the southern route, the reasons shaped the patterns of initial expansions of modern humans probably only limit to the discussions of the northern part of the Old World.

Response 2: We understand the suggestion, but we argue that our study covers much more than suggested here (see **Responses 1** and **14**).

3. When the authors said there are not enough data in the southern part of the Old world, it may due to the dating method the authors have chosen. Many of the sites presumably associated with early modern humans in those regions were dated to MIS 5 and 3 mainly by OSL and U-series methods. Some of the sites have been mentioned by the authors in lines 52-54.

Response 3: This is a valid point that we have now clarified in the main text (P6, L149–156):

“We acknowledge that our focus on the northern route ignores a potential southern route out of Africa to Asia through Ethiopia near the Red Sea⁴³ and the Arabian Peninsula⁴⁴, from where humans could have spread rapidly into regions of Southeast Asia and Oceania. However, many sites presumably associated with early modern humans in these regions are primarily dated using optically stimulated luminescence and uranium-thorium dating methods, which are not included in our database. Therefore, the inherent limitation of radiocarbon dating (~ 50,000 years ago⁴⁵) constrained our analysis to outside those regions (Extended Data Fig. 1).”

4. About the main results of this paper, there are a lot of room to discuss. The authors conclude that cultural drivers remain plausible at a finer scale, the migration corridors are predominantly constrained by regional environmental conditions. “Migration corridors are constrained by environmental conditions” seems like a common sense. It might be more important to learn why people moved at certain time from a region to another, and the various constrains among different regions. For instance, people arrived Far East at around 35 ka, but they moved forward largely after LGM. The common explanation is that the Bering land bridge hypothesis. However, LGM or other harsh environments may hinder the expansion of hominin groups, such as the refugia hypothesis in Europe. Migrations of modern humans from Levant to Europe are complicated as well when taking the earlier dispersals into account.

Response 4: We agree, and emphasise that we have an extensive discussion about the regional drivers of human expansion in *Supplementary Discussion 2* where we described the following examples:

1. The impact of precipitation and open spaces between the Fertile Crescent, Japan, and Beringia.

“The pathways connecting the Fertile Crescent with Japan and Beringia via Mongolia presented a similar pattern, with the routes circling north above Mongolia characterised by increased precipitation and either open spaces at the edges of forest or newly formed forests altogether; in contrast, humans travelling via more arid pathways south of Mongolia and crossing eastern Russia to reach Beringia (Fig. 1, and Extended Data Fig. 6a,d) benefited from the proximity of major rivers such as the Lena, Amur, and Yellow Rivers.”

2. The effect of river proximity and coastal limits on the pathways to Scandinavia and Western Europe.

“However, the impact of river proximity appeared to be more local in areas where climate conditions became more challenging (i.e., dry or forested habitats; Fig. 2a). For example, the pathways that led to Scandinavia and western Europe overall followed an increase in precipitation (Fig. 1 and Extended Data Fig. 6b) and forest-grassland ecotone habitats (Extended Data Fig. 6d). However, humans managed to cross some of the driest parts of these regions by following both the Danube and the Dnieper Rivers (Fig. 1 and Extended Data Fig. 6e), the former being consistent with archaeological evidence for this being a major conduit of human movement in the early Upper Palaeolithic^{30,31}. The differences in distributions of the Δ distances to

the nearest rivers (Fig. 2d) between the pathways leading to Scandinavia and the pathways leading to Western Europe show that in addition to following the Danube, humans also benefitted from coastal environments by following the route along the Mediterranean Sea (Figs. 1 and 2d).”

3. The pivotal role of river connectivity on the complex human expansion into South America.

“The expansion into South America is more complex because of the effort to avoid the generally dry conditions generated by the Antarctic Cold Reversal³⁷. The clockwise expansion pattern (Fig. 1) tracked the wettest regions of grasslands bordering the Amazonian Forest (Fig. 1, Extended Data Fig. 6b, d) along the east coast, as well as exploiting the connectivity provided by the Amazon, Paraguay, and Parana Rivers to cross more arid or forested regions and reach Chile (Fig. 1 and Extended Data Fig. 6e). The tributaries of the Amazon River (Fig. 1, and Extended Data Fig. 6e) led a westward expansion of Amazonian populations into the Andes despite the less-favourable climate conditions (drier and more forested; Fig. 1 and Extended Data Figs. 6b,d) compared to routes from the Pacific littoral or a trans-Altiplanic entry from the north³⁸.”

4. Following the reviewer’s suggestion, we also have expanded on the ‘Bering land bridge hypothesis’.

“Increased precipitation and the dominance of ecotone ecosystems were also essential components for humans to expand into the Americas, first entering along the western coast of North America ~ 16 ka. However, increasing temperatures from 17 ka³² led to an opening of the ice-free corridor in the Laurentide ice sheet 3 ky later³³, which created an additional path to North America by 12.6 ka³⁴ via the Mackenzie River (Figs. 1 and 3a). These changes in temperature provide the main explanation for the ~ 18.4-ky migration hiatus in Beringia before an initial entrance into North America at ~ 16.4 ka (Fig. 1)¹⁸ — almost 10 ky longer than previously suggested^{19,20}. This hiatus is referred to as the ‘standstill hypothesis’³⁵ or the ‘Beringian incubation model’²¹, and proposes that the ancestors of native Americans remained locally isolated because of ecological barriers such as the large ice sheets that covered North America at the time³⁶.”

We could not include this entire section in the main text because of the stringent word limit imposed by the journal. However, shall the editor agree to relax those limits, we would be happy to move most of this Supplementary Discussion back into the main text.

5. Lines 259-262. In the method part, I admire the efforts putting by the authors to sort out the reliable radiocarbon dates. However, the assumption that Middle Paleolithic (300-50 ka) was associated with Neanderthals and Upper Paleolithic (50-12 ka) was associated with modern humans was not clear. Did the author use dates as means to sort different hominin groups out, or both dates and cultures (Middle Paleolithic or Upper Paleolithic technology)? If only dates, it is not reasonable.

If both, it is better to make it clear. Denisovans in the Altai Mountain area and the edge of the Tibetan Plateau make things even more complicated.

Response 5: We indeed used both dates and cultures to differentiate the different hominin groups and made this point clearer (P11 L290–294):

“... we assumed that any material (i.e., dates and cultural-related technology) coming from Middle Palaeolithic layers (from 300,000 to 50,000 years ago) was associated with Neanderthals, whereas any material (i.e., dates and cultural related technology) from the Upper Palaeolithic (from 50,000 to 12,000 years ago) was associated with anatomically modern humans⁷¹.”

6. Line 271. Although I am not an expert studying genetics, the reason to exclude ancient DNA data is not easy to understand, at least based upon the current manuscript. And I am wondering whether the result will be different or not if including ancient DNA as we know the results indicated by the ancient DNA and by mitochondrial haplogroup are not always the same.

Response 6: We have now justified the reasons to exclude ancient DNA and why mtDNA and F_{ST} are best suited for our analysis (P11–12, L302–312).

“We gathered a total of 67,643 human mitochondrial control region sequences from Genebank (last accessed 28 May 2017, the last available update in the regions of interest). We excluded ancient DNA data because they are too scarce or even inexistant in most regions of the world to run short tandem repeat analyses and obtain enough nuclear coverage to address these movements of early modern humans. We then retained 27,506 sequences with reliable geographical locations to compute the fixation index (F_{ST}), a proven and robust method to infer genetic distance between populations⁷³.”

We agree with the referee that integrating ancient DNA datasets to have a temporal genetic framework would strengthen this analysis. Even though this is an impossible task given the scale of our analyses (due to data availability and processing time at a nearly global scale), we added this point as an element of discussion in the main text to highlight its potential benefit (P9, L236–242).

“While subject to the limitations of radiocarbon dating and the spatial distribution of reliably dated archaeological material, as well as assumptions of genetic distance between human populations, our results provide the most objective and robust representation of how and in what conditions humans dispersed so widely around the globe. Including ancient DNA data would provide an independent temporal genetic framework that would strengthen our analysis, but the scarcity of these data at a global scale and for the timeframe considered (i.e., the first human migrations) limits its application in the context of our study.”

7. Lines 378-379. Coastal movements would be the ones distant from the ocean? Distant or adjacent? When talking about the coastal movements, sea level fluctuations should be considered.

Response 7: We have clarified our definition of ‘coastal movement’ (P16, L430-431) as:

“... pathways with the shortest distance to the nearest ocean/sea ...”

We also re-ran our analyses accounting for sea level fluctuations and updated the method (P16–17, L434–437),

“We calculated the distance of each grid cell to its nearest coastline over the last 120 ky at a 1-ky interval. We accounted for the changes in sea level over time by iteratively applying a change of land-sea mask derived from a global sea-level record⁸⁶ overlaid onto present-day coastlines taken from the ETOPO1 dataset⁸⁷.”

Table 1 and *Supplementary Discussion 2*. Our results showed a slight increase in the importance of coastal movement (for some areas such as Beringia and Western Europe; see **response 19**), but it has not changed the overall message of our manuscript.

8. *The routes in Fig. 1 may need some explanations. For instance, Fig. 1 shows that modern humans crossed the Himalaya and Tibetan Plateau. It seems not a least cost way in many readers.*

Response 8: This is a valid point, but it seems to be a misunderstanding with the approach. We should have made it clearer in the text that we did not use a conventional least-cost path algorithm, but a modified least-cost path algorithm that we have adapted to infer human movements more realistically. We have now explained in the *Methods* (P14–15, L379–385) why a conventional least-cost path approach is not relevant for our study and why our algorithm does not always select the path of least cost/resistance:

“However, least-cost paths have been criticized because of the implicit assumption of a goal-oriented search between locations, i.e., that humans would take a single, optimal, least-cost path. This assumption would disregard any random exploratory behaviours and the likelihood that long-term, most-travelled corridors might not be the result of a single, least-cost path, but instead multiple pathways of lesser cost (‘least-cost’ corridors)⁸¹. By authorizing the selection of lesser goal-oriented pathways, some emergent ‘optimal’ trajectories might not always be the ‘least cost’ options.

and (P15, L406–409):

“This algorithm presents two main advantages compared to a classic least-cost path approach because it first introduces some randomness to mimic human behaviour (i.e., assuming that human decisions are not only based on cost-efficiency reasoning), and it guarantees a global spatial minimisation of pT (as opposed to a local minimisation of pT).

9. *A few minor comments. F column of the excel table is not countries.*

Response 9: Thank you for pointing this out; we have amended the table.

Reviewer #2

I could only read the manuscript briefly and not go in detail into the supplementary material (this was a caveat I highlighted when accepting the request to comment on this article), so please take my comments with a grain of salt.

I find the study interesting. From a brief skim it seems to me that the methods are well-explained and that some serious work has been put into this study. The results make a lot of sense, and it is nice to see a methodological, computational method applied to the assessment of this question, based on a large dataset that brings together genetic and carbon-dating approaches.

The genetic analysis seems pretty simplistic, but this may be an advantage at such a broad scale and with a lot of data.

I'm not sure how novel the findings are and whether the study really succeeds to achieve its goals. Some comments that perhaps the authors might be able to answer and could help in clarifying these questions:

1. The archaeological reconstruction is based on radiocarbon data? I'm no expert, but I think the limit of reliable inference of radiocarbon is ~40kya, and so founding on it the reconstruction of events that mostly happened earlier is odd (in other words: this should seemingly be reliable only in the Americas, and not in the spread most of the rest of the world, which happened earlier).

Response 10: We agree on the limitation of the radiocarbon dating, as we have pointed out in many places throughout the manuscript. But rather than a weakness, this aspect is in fact one of the main strengths of our approach — we combined the temporal and spatial distribution of the radiocarbon data into an innovative probabilistic framework (correcting for the Signor-Lipps effect) to infer timing of modern arrival sometimes even beyond the limit of reliable, direct inference possible from radiocarbon methods. We have now emphasized that point in the main text (P13–14, L351–355):

“By integrating both the temporal and spatial distributions of the radiocarbon dates and including their standard dating error into our probabilistic framework, we (i) correct for the Signor-Lipps effect in a spatially explicit way, and (ii) return reliable estimates of the timing of arrival at times even older than the limit of reliable, direct inference possible from radiocarbon methods.”

2. The dichotomy presented in the paper's narrative seems a bit like a false dichotomy: I don't think anyone suggests that environment didn't have a major influence on the spread of humans across the world. I find it hard to imagine a scenario in which the routes of human spread are not least-resistance routes across the landscape with respect to some environmental variable, regardless of whether cultural or other adaptation to local environments did or didn't occur in the process (or whether any other cultural dynamics, e.g. inter-group competition played a role, or cultural selection on dispersal, or within-group competition for resources, or anything like that. Even if such factors

were the main drivers of expansion, it would still have been expected to occur along routes that are more convenient in their environmental variables). The debate / open question might be (and is, I think) which environmental variables mattered, and which didn't (or more precisely: which more and which less).

Response 11: We have now toned down any unintended emphasis of this dichotomy (P2, L44–49):

“Human evolution and expansion over the last 120 ky (1 ky = 1000 years) are hypothesized to have been environmentally determined¹ because the occurrence of orbital-scale climate shifts^{2–4} might have created suitable habitat corridors for humans to exit Africa. However, cultural drivers, social structure, and interactions among different groups could have also explained movement patterns⁵.”

and we refocussed our main aim around the relative importance of environmental drivers (P3, L63–65):

“When focussing on how past environmental changes might explain this general pattern of human expansion, many climate-driven hypotheses are at the core of the plausible mechanisms, such as ...”

3. *If one really wants to tease apart the roles of cultural / population dynamics from the role of environmental constraints on spread, perhaps the way to go would have been (and is clearly not within the scope of the current study) checking whether the specific timing of bursts of spread is aligned with climatic fluctuations that would have opened/closed certain environmental corridors (such studies have / are being done; I'm not aware of the details, though). This is associated with many challenges, of course, and will also have its limitations.*

Response 12: We agree, but we are also confused because it was in a way (granted in a slightly different manner) what we had done in the previous version of this manuscript. In the previous version, we had stated in the *Methods* that we had calculated for each grid cell the climate anomalies (temperature and precipitation) at the specific timing of expansion (estimated by our model) relative to the present day. Climate anomalies more accurately describe climate variability over larger areas than do absolute temperatures (see ref 90), which is highly relevant to capture the spatio-temporal climate dynamic in the way the reviewer suggested. However, we did not extend this approach to the vegetation changes, instead we only looked at the percentage of forest and grassland along the different trajectories at the estimated timing of modern human arrival.

In the new version of this manuscript, we reran this analysis by more closely following the reviewer's suggestions, and we have recalculated for each grid cell the climate anomalies (i.e., temperature and precipitation) and the changes in vegetation (forest or grassland) at the specific timing of expansion (estimated by our model) relative to 90 ka (see justification in *Methods*). As the reviewer rightfully points out, this approach (adapted from ref 38) allowed us to capture the climate dynamic at the specific timing of human arrival, including the generation of new corridors of expansion. It is also worth mentioning that we are also now considering the change in sea level over time (see **Response 7**).

As shown in amended Table 1 in the main text, the new results slightly changed the final probability for each environmental variable, but the regional ranking remained the same. We have also amended the relevant section of the *Methods* (P17, L443–446):

“To capture the broad-scale climate dynamic at the time of human expansion⁹⁰, we calculated the values of mean annual temperature, and mean annual precipitation at the estimated timing of arrival in each grid cell (Extended Data Fig. 4a) and relative to 90 ka (see Extended Data Fig. 6a, b respectively).

and (P17, L457–462)

“We used the global simulations of vegetation changes over the last 120 ky from the BIOME4 vegetation model forced offline using the HadCM3 climate simulation (see details in Supplementary Method 2)^{88,92} and calculated the changes in simulated dominant vegetation type (forest or grassland biome) at the estimated timing of arrival in each grid cell (Extended Data Fig. 4a) and relative to 90 ka (see Extended Data Fig. 6d).”

and we also amended accordingly the maps of changes for each climate variable at the time of arrival that were provided in *Extended Data Fig.6*.

4. Continuing from (2), one would expect the study to explore a large number of environmental variables and highlight those that were more/less important; but (again, my reading was not as thorough as I'd like, I apologize if I'm wrong here) it seems that the conclusion is that pretty much all/most of the variables that were explored turned out to be important (perhaps because they are inevitably non-independent of one another).

Response 13: We agree, which is what we tried to do while avoiding the problem of multicollinearity (i.e., whenever an independent variable is highly correlated ≥ 1 of the other independent variables) between the environmental variables that would undermine the strength of our results.

Indeed, the other environmental variables provided by the HadCM3 model are mostly related to annual temperatures and precipitation, e.g., seasonal temperature and precipitation, or temperature and precipitation of the warmest (July) and coldest month (January). Adding these variables would only make the problem of multicollinearity more prevalent. We have now clarified this point in the *Methods* (P17, L439–443).

“We used the already published mean annual temperature and precipitation simulations for the last 120 ky from the HadCM3 Atmosphere-Ocean General Circulation Model⁸⁸. The HadCM3 climate model also provides monthly and seasonal temperatures and precipitation, but we excluded these variables from the analysis to avoid multicollinearity effects⁸⁹.”

However, we disagree with this statement: “... the conclusion is that pretty much all/most of the variables that were explored turned out to be important”. We clearly showed in our analysis that only three of the environmental variables explain most of the pattern of modern human expansion. Most importantly, we show that their relative importance varies regionally

as detailed in **Response 4**, which represents the true novelty and originality of our study (also see **Response 15**).

5. Furthermore, there's been a major debate about the role of coastal expansion and a southern route of spread (Arabia, south Asia), but the data doesn't allow to test this, so a major aspect of the debate about the role of environmental adaptation to facilitating/hindering spread could not be addressed.

Response 14: This is a valid point, and we have now addressed this issue in the *Supplementary Method, Supplementary Discussion* and the main text:

1. We added an extra section in '*1. Inferring the regional timing of initial human arrival*' detailing the regions that we excluded due to the ~ 50-ka limitation of radiocarbon dating:

“The ~ 50-ka limitation to radiocarbon dating¹ prevented estimating arrival times into the Levant and the Arabian Peninsula more generally during Marine Isotope Stage 5 (126–74 ka), and the region connecting Africa to Australia²⁻⁴. We therefore excluded the following countries in our study that are primarily dated using optically stimulated luminescence and uranium-thorium methods: Saudi Arabia (2,149,690 km²), Yemen (527,968 km²), Oman (309,500 km²), United Arab Emirates (83,600 km²), Pakistan (881,912 km²), India (3,287,590 km²), Nepal (147,181 km²), Bangladesh (147,570 km²), Myanmar (676,578 km²), Laos (236,800 km²), Thailand (513,212 km²), Vietnam (331,212 km²), Cambodia (181,035 km²), Malaysia (330,803 km²), Indonesia (1,904,569 km²), and Philippines (342,353 km²). We therefore restricted our focus to the ‘northern route’ of expansion via the Fertile Crescent, rather than the ‘southern route’ following the Strait of Bab-el-Mandab⁵.”

2. We added an extra section in '*2. Regional environmental drivers of human expansion*' to discuss the consequences of this choice on our results. We acknowledge that the debate about the role of coastal expansion and a southern route of spread is out of the scope of our study, but we also pointed out that the regions we excluded represent < 10% of the total landmass of our study area, making our results regarding the environmental drivers of modern human expansion (including coastal changes) still highly relevant.

“We acknowledge that by focusing on the Eurasian ‘northern route’ of expansion via the Fertile Crescent (mostly due to the limitation of the dating method), we cannot address the major debate about the relative roles of coastal expansion and environmental adaptation along the southern spread route in Eurasian (via Arabia and south Asia). However, the total land area we excluded is ~ 8,980,723 km² (i.e., sum of the areas of each excluded country described in Supplementary Method 1), which represents ~ 9.5% of our entire study area (i.e., Eurasia + America = ~ 94,000,000 km²). We therefore argue that despite the importance of the south Eurasian route of expansion, our results still show the role of environmental adaptations (including coastal changes such as in the Pacific Northwest and northern Europe) to facilitate/hinder the spread of modern humans across > 90% of both Eurasia and the Americas combined.”

3. As detailed in **Response 7**, we re-ran our analysis accounting for the changes in coastline over time, and expanded on the importance of the distance to coast in some regions.

In sum – I like this study, but I’m not convinced that it is a breakthrough in our understanding of human spread, and the novelty isn’t completely clear to me (beyond the fact that it was done in what seems to be a methodical approach with orderly and systematic treatment of a large dataset, which are important).

Perhaps a different pitch or some clarifications could settle these open questions, but maybe not – I’m sincerely not sure. I’m sure the study will be published, and I’ll be happy to see it in the literature; I’m not sure whether Nat Comms is the appropriate venue or not. (I mean this literally; I’m not saying it isn’t, I’m saying I can’t tell whether it is or is not).

Response 15: We are of course delighted with the support, but disagree that *Nature Communications* is not the right venue based on a novelty argument. We have now made it clear in the main text that this is the first study of its kind to investigate the spread of humans by statistically combining genetic data, archaeology, and climate modelling into a continuous spatial framework.

Without our framework, the interpretation of the main drivers of modern human expansion were only based on weak (non-quantitative inference) from scattered, independent local events. On the contrary, we overcame the lack of spatially and temporally continuous data (from other approaches) to show how environmental determinants are intertwined and how the dominance of one factor relative to others changes regionally. We have now clarified this point in the main text (P9, L242–247):

“More importantly, we have provided an innovative and robust approach that for the first time quantitatively combines genetics, archaeology, and climate modelling into a continuous spatial framework. This framework builds on the strength of each discipline while overcoming their inherent limitations to test scenarios that would not be possible to evaluate by relying solely on sparse and incomplete ‘snapshot’ data of local and spatially isolated events.”

Line 169: The “incompatible” seems like it isn’t in place.

Response 16: We have rewritten this sentence by removing the double negative to make it clearer (P7, L173–174):

“... our results are consistent with expeditions of foragers from the Pacific littoral zone expanding their access to resources^{48,53}.”

Reviewer #4

Saltre et al used archaeological and genetic data to predict the most likely path taken by modern humans to be spread in Eurasia and America. The path they predicted broadly agrees with what is

already known by genetic data, but combining with archaeological data, they can make more robust inferences. Furthermore, by looking at the environmental variables of these paths, they can predict what might be the essential factor for human dispersal. Although the methods used here are fascinating, I have some points before it is acceptable for this journal.

Major:

1. The authors said they have only used modern mitochondrial data for this analysis. However, as we know that there are several substantial population replacements of indigenous populations (at least in Europe and Middle East) I am not sure that using only modern data is enough for mitochondria. Especially as the authors deliberately removed all the ancient genome, which might be a better representative of the actual ancient mitochondrial status of the region rather than the modern population. Authors should either use ancient genomes or discuss why ancient genomes were removed in this case.

Response 17: We have addressed this point in **Response 6**.

2. Although Figure 1 is generally quite good in representing the human out of Africa migration with appropriate time points, I have noticed some results are a little bit debatable. Especially a human population was migrating from Central Asia to Japan (close to 10,000 km) in less than 1,500 years. The authors should discuss some possible drawbacks of such a fast expansion rate (for example, underestimation of time in Central Asia due to unavailability of archaeological data). A supplementary Table will be appreciated with the distance covered and time taken for different paths (Africa to Beringia, Beringia to Mexico etc.). Also, some citations about the expansion rate and if the authors' results match that.

Response 18: As suggested, we have added Supplementary Table 3 that includes the distance covered, the time taken, and the required speed to travel all the different pathways (both *likely routes* and *less-likely routes*). We also have added to the Discussion an extra paragraph contextualising the expansion rate relative to previously published values (P7–8, L175–194):

“Our estimated migration rates (Extended Table S3) are higher than those reconstructed for Sahul (0.71–0.92 km year⁻¹)⁵⁴ or European Neolithic farmers (~ 1 km year⁻¹)^{55,56}, but our results remain plausible. First, our estimates represent the spread of hunter-gatherers that differ from farmers, because the development of farming technology suppresses the expansion rates of more sedentary agriculturalists⁵⁷. Second, human movement depends on the perception of resource availability (or depletion) triggering their next move⁵⁸, which often results in either frequent, short-distance movements across warm, highly productive environments, or infrequent, long-distances movements in low-productivity environments. At a continental scale, and assuming humans followed ecosystems of forest-grassland transition (Fig. 1 and 2c), we expect an intermediate strategy of frequent, long-distance movements that would support our results. Third, our estimated expansion rates across western Europe and North America (Extended Table S3) match some of those derived from modern-day hunter-gatherer societies that move every three weeks at a highly variable daily travel distance (0.4–15 km year⁻¹) depending on the landscape (on/off natural trails)^{59,60}. Finally, some of the fastest expansion rates (e.g., across Asia and South America; Extended Table S3) could have resulted from some

methodological limitations caused by a lack of data in those areas (Extended Fig. data 1a). An increase in the distribution of available data in those areas would help to refine our local-scale inferences and likely decrease the estimated travel velocity (even though some rare movements of 60–80 km day⁻¹ have been recorded^{61,62})(Extended Table S3).”

3. In Page 8 line 206: "such as in the driest part of Scandinavia and Western Europe via both the Danube and the Dnieper Rivers": I am not sure why the mean and confidence interval of Scandinavia and Western Europe shows two different directions for the nearest river. Whereas Western Europe shows a positive tendency, Scandinavia shows a negative tendency. Can the author explain it better for the opposite results?

Response 19: This is a good point. We have now explained in the *Supporting Discussion 2* that the main differences between these pathways is that the those leading to Scandinavia are inland (only relying on river connectivity to travel across regions with challenging climate conditions), whereas the pathways leading to western Europe also benefitted from the coastal environment along the Mediterranean Sea:

“However, the impact of river proximity appeared to be more local in areas where climate conditions became more challenging (i.e., dry or forested habitats; Fig. 2a). For example, the pathways that led to Scandinavia and western Europe overall followed an increase in precipitation (Fig. 1 and Extended Data Fig. 6b) and forest-grassland ecotone habitats (Extended Data Fig. 6d). However, humans managed to cross some of the driest parts of these regions by following both the Danube and the Dnieper Rivers (Fig. 1 and Extended Data Fig. 6e), the former being consistent with archaeological evidence for this major conduit of human movement in the early Upper Palaeolithic^{30,31}. The differences in distributions of the Δ distances to the nearest rivers (Fig. 2d) between the pathways leading to Scandinavia and those leading to western Europe show that in addition to following the Danube, humans also benefitted from coastal environments by following the route along the Mediterranean Sea (Figs. 1 and 2d).”

Minor:

1. The F_{ST} calculation is not clear for mitochondrial data. Are they directly calculated by the distance between the mitochondrial genome? Or are they calculated from the known distance coming from haplogroup information?

Response #20: We have now expanded the description of the F_{ST} calculation (P12, L308–317):

“Mitochondrial haplogroup were classified from control region sequences using HaploGrep 2⁷⁴, according to nomenclature provided by PhyloTree mtDNA tree Build 17 (phyloree.org; ref 75), giving a total of 31 haplogroups (A, B, C, D, E, F, G, H, I, J, K, L0, L1, L2, L3, L4, L5, M, N, O, P, Q, R, S, T, U, V, W, X, Y, Z). We sorted the data by geographical location by grouping the samples according to 96

geographical locations, which were either country or region/province. We then converted the data into Genpop format using the R package `pegas` and generated the matrix of pairwise F_{ST} between the 96 geographical locations using haplogroup frequencies⁷⁶ and the R package `diveRsity`. Ultimately, we removed non-indigenous haplogroups.”

2. Similarly, in case there are several individuals from the same regions, how the F_{ST} is calculated for them?

Response 21: See **Response 20**.

3. Extended Figure 1b: the dots, if they represent mitochondrial DNA samples, do not look like there are 27 thousand samples. Please clarify. Also, a better description is needed for the caption.

Response 22: We have now expanded the caption of the Extended Data Figure 1b, making it clear that the red dots represent the centroids of the 96 geographical locations described in Response 20:

“Spatial distribution of **a.** reliable, high-quality (see *Methods*), radiocarbon-dated archaeological specimens indicating human presence (orange dots), and **b.** mitochondrial DNA samples (red dots) used to generate the map of F_{ST} pairwise distances ranging from light green (short F_{ST} pairwise distances) to dark blue (long F_{ST} pairwise distances). The red dots represent the centroids (size of each centroid is indicated by the colour bar of F_{ST} pairwise distances) of the 96 geographical locations (either country or region/province) used to group the 31 mitochondrial haplogroups compiled from the 67,643 human mitochondrial control region sequences from Genbank.”

4. Is there any particular reason to use such an older dataset for Genbank (28 May 2017)?

Response 23: We have now clarified that this date was the last available update in the regions of interest (P11, L302–303).

“We gathered a total of 67,643 human mitochondrial control region sequences from Genbank (last accessed 28 May 2017, the last available update in the regions of interest).”

5. It would be better to have a little bit more time data point in Figure 1. For example, the separation time between European and East Asian in the Middle East (the yellow dot).

Response 24: Thank you for this useful suggestion. We have now added five extra time data in the recommended area (see new Figure 1). Adding even more points would reduce the quality of the figure, but we hope that we are now displaying enough dates to provide the reader with a good overview of the patterns of modern human movements.

6. Page 8 line 201: *Not sure what is meant by Fig. 3c. I can see only two main figures.*

Response 25: Apologies, we meant Fig. 2c, this is now corrected in P9, L224–227.

“The most-likely routes of human migration were also located mainly on (newly) opened landscapes (Fig. 2c), which indicates that humans travelled through grassland areas while staying close to ancient or newly formed forests (Figs. 1, 2c and Extended Data Fig. 6d).”

7. *Figure 2: It would be great if the authors could sort the median and confidence interval accordingly, as it is mentioned in the footnote. Easy to find which colour corresponds to which path.*

Response 26: We have sorted the medians and the confidence intervals accordingly.

8. *Page 8 line 207: (i.e., dry or forested habitats; Fig. 2a). I thought Figure 2a represented temperature, not dryness.*

Response 27: We have now amended the text with the correctly referring to Fig. 2b (for dryness) and Fig 2c (for forested habitats).

Reviewers' Comments:

Reviewer #1:

Environmental conditions shaped the patterns of initial expansion by anatomically modern humans

Understanding the peopling of various continents and regions of our species is for sure an interesting and also a key topic in the studies of human evolution. Archaeological evidence is accumulative which means the earliest evidence of modern human presence in a certain region updates quickly through time. Modeling then becomes a complementary and sometimes efficient way to understand the dispersal of modern humans. In the manuscript, Dr. Saltré et al compiled a huge number of radiocarbon dates and genetic data, and developed a statistical approach to indicate the most-likely initial expansion routes of modern humans in Eurasia and the Americas. The results provide us a new scenario understanding the dispersal of early modern humans in the northern part of the old world and the new world. However, we should also be aware of that those routes are more likely presumptive for that it works only if the archaeological and genetic data are both complete and accurate which may not be always the case. And this shortage of archaeological data has been noticed by the authors.

I am sure there are other reviewers who will check the mathematics and statistics of the simulation. Here I will comment on the archaeology mainly.

1. The title of the manuscript may not be appropriate for that the authors mainly discussed the northern route of the dispersal, while the southern routes which are more familiar by many readers are not included.

Response 1: This is a valid point. We tried to make the title as general as possible because although it is correct that we excluded the southern routes, those routes only represent < 10% of the total land mass of our study area. We have now addressed this point in detail in the Supplementary Methods, Supplementary Discussion, and the main text (see Response 14 for more details).

Reply from Referee: It is better now, and at least the readers know the reasons.

2. The following comment relates to the first one. Without the data of the southern route, the reasons shaped the patterns of initial expansions of modern humans probably only limit to the discussions of the northern part of the Old World.

Response 2: We understand the suggestion, but we argue that our study covers much more than suggested here (see Responses 1 and 14).

3. When the authors said there are not enough data in the southern part of the Old world, it may due to the dating method the authors have chosen. Many of the sites presumably associated with early modern humans in those regions were dated to MIS 5 and 3 mainly by OSL and U-series methods. Some of the sites have been mentioned by the authors in lines 52-54.

Response 3: This is a valid point that we have now clarified in the main text (P6, L149–156): “We acknowledge that our focus on the northern route ignores a potential southern route out of Africa to Asia through Ethiopia near the Red Sea⁴³ and the Arabian Peninsula⁴⁴, from where humans could have spread rapidly into regions of Southeast Asia and Oceania. However, many sites presumably associated with early modern humans in these regions are primarily dated using optically stimulated luminescence and uranium-thorium dating methods, which are not included in our database. Therefore, the inherent limitation of radiocarbon dating (~ 50,000 years ago⁴⁵) constrained our analysis to outside those regions (Extended Data Fig. 1).”

Reply from Referee: The authors have made this point clear.

4. About the main results of this paper, there are a lot of room to discuss. The authors conclude that cultural drivers remain plausible at a finer scale, the migration corridors are predominantly constrained by regional environmental conditions. "Migration corridors are constrained by environmental conditions" seems like a common sense. It might be more important to learn why people moved at certain time from a region to another, and the various constrains among different regions. For instance, people arrived Far East at around 35 ka, but they moved forward largely after LGM. The common explanation is that the Bering land bridge hypothesis. However, LGM or other harsh environments may hinder the expansion of hominin groups, such as the refugia hypothesis in Europe. Migrations of modern humans from Levant to Europe are complicated as well when taking the earlier dispersals into account.

Response 4: We agree, and emphasise that we have an extensive discussion about the regional drivers of human expansion in Supplementary Discussion 2 where we described the following examples:

1. The impact of precipitation and open spaces between the Fertile Crescent, Japan, and Beringia.

"The pathways connecting the Fertile Crescent with Japan and Beringia via Mongolia presented a similar pattern, with the routes circling north above Mongolia characterised by increased precipitation and either open spaces at the edges of forest or newly formed forests altogether; in contrast, humans travelling via more arid pathways south of Mongolia and crossing eastern Russia to reach Beringia (Fig. 1, and Extended Data Fig. 6a,d) benefited from the proximity of major rivers such as the Lena, Amur, and Yellow Rivers."

2. The effect of river proximity and coastal limits on the pathways to Scandinavia and Western Europe.

"However, the impact of river proximity appeared to be more local in areas where climate conditions became more challenging (i.e., dry or forested habitats; Fig. 2a). For example, the pathways that led to Scandinavia and western Europe overall followed an increase in precipitation (Fig. 1 and Extended Data Fig. 6b) and forest-grassland ecotone habitats (Extended Data Fig. 6d). However, humans managed to cross some of the driest parts of these regions by following both the Danube and the Dnieper Rivers (Fig. 1 and Extended Data Fig. 6e), the former being consistent with archaeological evidence for this being a major conduit of human movement in the early Upper Palaeolithic^{30,31}. The differences in distributions of the Δ distances to the nearest rivers (Fig. 2d) between the pathways leading to Scandinavia and the pathways leading to Western Europe show that in addition to following the Danube, humans also benefitted from coastal environments by following the route along the Mediterranean Sea (Figs. 1 and 2d)."

3. The pivotal role of river connectivity on the complex human expansion into South America.

"The expansion into South America is more complex because of the effort to avoid the generally dry conditions generated by the Antarctic Cold Reversal³⁷. The clockwise expansion pattern (Fig. 1) tracked the wettest regions of grasslands bordering the Amazonian Forest (Fig. 1, Extended Data Fig. 6b, d) along the east coast, as well as exploiting the connectivity provided by the Amazon, Paraguay, and Parana Rivers to cross more arid or forested regions and reach Chile (Fig. 1 and Extended Data Fig. 6e). The tributaries of the Amazon River (Fig. 1, and Extended Data Fig. 6e) led a westward expansion of Amazonian populations into the Andes despite the less-

favourable climate conditions (drier and more forested; Fig. 1 and Extended Data Figs. 6b,d) compared to routes from the Pacific littoral or a trans-Altiplanic entry from the north³⁸.”

4. Following the reviewer’s suggestion, we also have expanded on the ‘Bering land bridge hypothesis’.

“Increased precipitation and the dominance of ecotone ecosystems were also essential components for humans to expand into the Americas, first entering along the western coast of North America ~ 16 ka. However, increasing temperatures from 17 ka³² led to an opening of the ice-free corridor in the Laurentide ice sheet 3 ky later³³, which created an additional path to North America by 12.6 ka³⁴ via the Mackenzie River (Figs. 1 and 3a). These changes in temperature provide the main explanation for the ~ 18.4-ky migration hiatus in Beringia before an initial entrance into North America at ~ 16.4 ka (Fig. 1) ¹⁸ — almost 10 ky longer than previously suggested ^{19,20}. This hiatus is referred to as the ‘standstill hypothesis’ ³⁵ or the ‘Beringian incubation model’ ²¹, and proposes that the ancestors of native Americans remained locally isolated because of ecological barriers such as the large ice sheets that covered North America at the time ³⁶.”

We could not include this entire section in the main text because of the stringent word limit imposed by the journal. However, shall the editor agree to relax those limits, we would be happy to move most of this Supplementary Discussion back into the main text.

Reply from Referee: As the authors explained environmental conditions shaped human migration are quite diverse from regions to regions. It could be better to make this clear in the main text, however the discussion is limited in the main text now. There are much vital information only showed in the Supplementary material, and this is for sure hinder the readers fully understanding the paper.

5. Lines 259-262. *In the method part, I admire the efforts putting by the authors to sort out the reliable radiocarbon dates. However, the assumption that Middle Paleolithic (300-50 ka) was associated with Neanderthals and Upper Paleolithic (50-12 ka) was associated with modern humans was not clear. Did the author use dates as means to sort different hominin groups out, or both dates and cultures (Middle Paleolithic or Upper Paleolithic technology)? If only dates, it is not reasonable. If both, it is better to make it clear. Denisovans in the Altai Mountain area and the edge of the Tibetan Plateau make things even more complicated.*

Response 5: We indeed used both dates and cultures to differentiate the different hominin groups and made this point clearer (P11 L290–294):

“... we assumed that any material (i.e., dates and cultural-related technology) coming from Middle Palaeolithic layers (from 300,000 to 50,000 years ago) was associated with Neanderthals, whereas any material (i.e., dates and cultural related technology) from the Upper Palaeolithic (from 50,000 to 12,000 years ago) was associated with anatomically modern humans⁷¹.’

Reply from Referee: The authors should address the Denisovan issue. Are they excluded? And why?

6. Line 271. *Although I am not an expert studying genetics, the reason to exclude ancient DNA data is not easy to understand, at least based upon the current manuscript. And I am wondering whether the result will be different or not if including ancient DNA as we know the results indicated by the ancient DNA and by mitochondrial haplogroup are not always the same.*

Response 6: We have now justified the reasons to exclude ancient DNA and why mtDNA and FST are best suited for our analysis (P11–12, L302–312).

“We gathered a total of 67,643 human mitochondrial control region sequences from Genbank (last accessed 28 May 2017, the last available update in the regions of interest). We excluded ancient DNA data because they are too scarce or even inexistant in most regions of the world to run short tandem repeat analyses and obtain enough nuclear coverage to address these movements of early modern humans. We then retained 27,506 sequences with reliable geographical locations to compute the fixation index (FST), a proven and robust method to infer genetic distance between populations⁷³.”

We agree with the referee that integrating ancient DNA datasets to have a temporal genetic framework would strengthen this analysis. Even though this is an impossible task given the scale of our analyses (due to data availability and processing time at a nearly global scale), we added this point as an element of discussion in the main text to highlight its potential benefit (P9, L236–242).

“While subject to the limitations of radiocarbon dating and the spatial distribution of reliably dated archaeological material, as well as assumptions of genetic distance between human populations, our results provide the most objective and robust representation of how and in what conditions humans dispersed so widely around the globe. Including ancient DNA data would provide an independent temporal genetic framework that would strengthen our analysis, but the scarcity of these data at a global scale and for the timeframe considered (i.e., the first human migrations) limits its application in the context of our study.”

Reply from Referee: It is reasonable.

7. Lines 378-379. Coastal movements would be the ones distant from the ocean? Distant or adjacent? When talking about the coastal movements, sea level fluctuations should be considered.

Response 7: We have clarified our definition of ‘coastal movement’ (P16, L430-431) as: “... pathways with the shortest distance to the nearest ocean/sea ...” We also re-ran our analyses accounting for sea level fluctuations and updated the method (P16–17, L434–437), “We calculated the distance of each grid cell to its nearest coastline over the last 120 ky at a 1-ky interval. We accounted for the changes in sea level over time by iteratively applying a change of land-sea mask derived from a global sea-level record⁸⁶ overlaid onto present-day coastlines taken from the ETOPO1 dataset⁸⁷.” Table 1 and Supplementary Discussion 2. Our results showed a slight increase in the importance of coastal movement (for some areas such as Beringia and Western Europe; see response 19), but it has not changed the overall message of our manuscript.

Reply from Referee: Clear now.

8. The routes in Fig. 1 may need some explanations. For instance, Fig. 1 shows that modern humans crossed the Himalaya and Tibetan Plateau. It seems not a least cost way in many readers.

Response 8: This is a valid point, but it seems to be a misunderstanding with the approach. We should have made it clearer in the text that we did not use a conventional least-cost path algorithm, but a modified least-cost path algorithm that we have adapted to infer human movements more realistically. We have now explained in the Methods (P14–15, L379–385) why a conventional least-cost path approach is not relevant for our study and why our algorithm does not always select the path of least cost/resistance:

“However, least-cost paths have been criticized because of the implicit assumption of a goal-oriented search between locations, i.e., that humans would take a single, optimal, least-cost path. This assumption would disregard any random exploratory behaviours and the likelihood that long-term, most-travelled corridors might not be the result of a single, least-cost path, but instead multiple pathways of lesser cost (‘least-cost’ corridors)⁸¹. By authorizing the selection of lesser goal-oriented pathways, some emergent ‘optimal’ trajectories might not always be the ‘least cost’ options.

and (P15, L406–409):

“This algorithm presents two main advantages compared to a classic least-cost path approach because it first introduces some randomness to mimic human behaviour (i.e., assuming that human decisions are not only based on cost-efficiency reasoning), and it guarantees a global spatial minimisation of pT (as opposed to a local minimisation of pT).

Reply from Referee: The approach used by the authors makes sense, but my concern is the results. Crossing Himalaya and Tibetan Plateau right in the middle instead of the edge is a surprising result, and this needs explanations.

9. A few minor comments. F column of the excel table is not countries.

Response 9: Thank you for pointing this out; we have amended the table.

Reviewer #2:

Remarks to the Author:

The authors have addressed my comments in the previous round of review. Although I remain with some reservations about the novelty of the findings and the ability of this data to account for some of the to-be-explained observations (e.g. the limit of radiocarbon data to 40K, while the processes being explained primarily took place earlier than that time), I think the study is valuable, and I don't feel that further back-and-forth will improve much.

I leave to the other reviewers to determine whether this study represents a novel-enough breakthrough in order to be published in Nat Comms, as I am undecided regarding this point.

I believe further emphasis in the discussion on the limitations of the methods to account for the phases prior to 40KYA would be useful, and a clearer statement (with refs) about what precisely has been done already and where the current study's greatest novelty is, would perhaps help the reader appreciate the manuscript better.

Reviewer #4:

Remarks to the Author:

Authors have improved the manuscript a lot by answering my questions but I still need some clarifications:

1. I think the Table 1 with inclusion of new results makes less sense. A better caption is needed. In this regard, it is difficult to fathom why "d coast" and "d river" is higher for most likely paths compared to less likely. This is not intuitive at all (assumption would be modern humans follow the path where either of these distances are lower not higher) and needs some discussion in the manuscript unless I am misunderstanding something.

2. Page 5 Line 125: "We extracted the genetic dataset from a GenBank search query to find 67,643 human mitochondrial control-region sequences, cleaned of ancient DNA data and any non-Indigenous lineages.": the number of samples (67,643) seems misleading as in the end 27,506 samples were used. Either mention both the numbers or mention the final sample number that was used to calculate the Fst.

3. I am a little bit confused about how the environmental variables are calculated. For example: Page 16, Line 421: mean annual temperature anomaly (at the estimated timing of arrival; Extended Data Fig. 4a) relative to the present .

Page 19, Line: mean annual precipitation relative to 90 ka.

Page 4, line 97: after they exited Africa over 65 ka.

Is it relative to the present or it is relative to 90ka which is used for the calculation? Is it indeed 90 ka which is used for the Out of Africa event, as 65 ka was mentioned also? The authors should follow one parameter throughout the paper to avoid confusion. Specially the parameter in this case should be 65ka as the authors are not concentrating on the coastal path of out of Africa migration.

Reviewer #5:

Remarks to the Author:

I have been asked to consider Reviewer 1's comments and recommendations, which I have done. I have also reviewed comments from the other reviewers, some of which echo Reviewer 1's concerns.

With respect to Reviewer 1 (Comments 1,2 Responses 1,2): The authors admit that they ignore the southern route out of Africa. There are a couple of concerns here. The first is that the southern route has been the subject of numerous publications, including numerous high impact publications, over two decades. To ignore the southern route seems entirely implausible to me given that the authors aim to examine the "initial expansion of anatomically modern humans". Even if the authors believe the

southern route played no role in the dispersal of our species, the complete absence of any data or analysis is a concern.

With respect to Reviewer 1 (Comment 3, Response 3): The second problem related to the above, is the fact that the authors ignore all sites and dates with luminescence and uranium series methods. I find this to be unreasonable. Indeed, I find the study methodologically flawed from the outset if it relies only on radiocarbon dates for examining human fossils and archaeological sites associated with anatomically modern humans (essentially the same point as Reviewer 2, Comment 1).

With respect to Reviewer 1 (Comment 5, Response 5): The assumption that the Middle Palaeolithic is associated with Neanderthals is flawed. There are clear associations of *Homo sapiens* with Middle Palaeolithic technologies outside of Africa! Please remember that the majority of the evolution of *Homo sapiens* in Africa was associated with the Middle Stone Age, so any dispersals prior to the Later Stone Age would be associated with Middle Palaeolithic toolkits. There are a number of publications which demonstrate this point.

Environmental conditions shaped the patterns of initial northern expansion of anatomically modern humans

Reviewer #1

1. As the authors explained environmental conditions shaped human migration are quite diverse from regions to regions. It could be better to make this clear in the main text, however the discussion is limited in the main text now. There are much vital information only showed in the Supplementary material, and this is for sure hinder the readers fully understanding the paper.

Response #1: The suggested change strengthens the manuscript and with the editor's approval (in terms of word limit), we have now included in the main text the section of the supplementary material describing regional variation in environmental drivers of human expansion (P8–11, L218–279).

“However, these environmental variables are intertwined and the dominance of one factor relative to others changes regionally (Fig. 1). Despite the continental decrease in temperature (Extended Data Fig. 7), ... , compared to routes from the Pacific littoral or a trans-Andean entry from the north⁵⁰.”

2. The authors should address the Denisovan issue. Are they excluded? And why?

Response #2: Addressing Denisovan (or Neanderthal, or any other *Homo* spp.) movements was never the subject or scope of our study — we are focussing exclusively on the environmental drivers of the northern expansion of *Homo sapiens*. Including these species would have required accounting for demographic processes and interactions among species. While this might be valuable, it would entail an entirely different modelling approach and data (e.g., fertility and survival rates, nature of interactions, etc.) that are not available at such a spatiotemporal scale. We have clarified this on P12, L302–306.

“Including Denisovan and/or Neanderthal interactions could potentially be important determinants of the expansion patterns of *Homo sapiens*, but such considerations would require demographic data for each species (e.g., fertility and survival) and the nature of the interactions among species (e.g., warfare, competition for resources, interbreeding, etc.) that are not available at the spatiotemporal scale we were able to model in this study.”

3. The approach used by the authors makes sense, but my concern is the results. Crossing Himalaya and Tibetan Plateau right in the middle instead of the edge is a surprising result, and this needs explanations.

Response #3: We have now expanded discussion on this point (P9, L235–241).

“Our results did not show major differences in the ruggedness and the distance to the nearest river between the most-likely routes of human expansion and the least-likely routes, except in some parts of South America (Extended Data Fig. 6d). This lack of effect of the ruggedness explains one of the most likely routes crossing the Himalaya and Tibetan Plateau in the middle (Fig. 1) as the fastest way to reach the 4,600 m above sea level Nwya Devu in the interior of the Tibetan plateau as early as 40,000 years ago as supported by archaeological evidence⁶⁶.”

Reviewer #2

1. I believe further emphasis in the discussion on the limitations of the methods to account for the phases prior to 40KYA would be useful, and a clearer statement (with refs) about what precisely has been done already and where the current study's greatest novelty is, would perhaps help the reader appreciate the manuscript better.

Response #4: We have now expanded the discussion following these recommendations (P11, L288–299).

“More importantly, we have provided an innovative and robust approach that for the first time quantitatively combines genetics, archaeology, and climate modelling into a continuous spatial framework rather relying on one single type of data or method. Our framework builds on the strength of each discipline while overcoming their inherent limitations to test scenarios that would not be possible to evaluate by relying solely on sparse and incomplete ‘snapshot’ data of local and spatially isolated events. Moreover, our statistical approach to infer spatio-temporal trajectories of initial human expansion overcomes the methodological limitations of most previous contributions in this area such as: (i) the under-representation of older arrival events⁷⁵ and the Signor-Lipps effect³⁵, (ii) either not spatially explicit⁷⁶, or (iii) generating new spatial biases via arbitrary geographic binning^{27,77,78}, or when interpolating a linear chronology from unevenly spaced age estimates⁷⁹, and/or (iv) neglecting uncertainty arising from sampling and taphonomic biases⁸⁰, and inherent dating error⁸¹.”

Reviewer #4

1. I think the Table 1 with inclusion of new results makes less sense. A better caption is needed. In this regard, it is difficult to fathom why “d coast” and “d river” is higher for most likely paths compared to less likely. This is not intuitive at all (assumption would be modern humans follow the path where either of these distances are lower not higher) and needs some discussion in the manuscript unless I am misunderstanding something.

Response #5: We apologise for the confusion and have amended the caption (P35, L904–914) to match the description in the *Methods* (P20, L520–530) and clarify the two following points:

- 1) **mean ± sd** are summary statistics of the environmental variable calculated across all continents. While the **mean** values of *d coast* and *d river* for the most likely paths are

higher than less-likely ones, their high associated **sd** make these differences meaningless. This is because calculations at a continental scale masked any regional variation, so that the most-likely routes in areas driven by low distance to coast or river would be cancelled out by most-likely route in other areas driven by another environmental variable and presenting higher values of distance to coast or river.

- 2) **prob** fixed this issue by accounting for these regional differences because each probability is calculated by standardising the average value of each environmental variable along each route by its length.

“**Table 1** – Estimated probability (**prob** = $1 - P_{\text{rand}}$) that the difference between environmental values (**env variable**) along the most-likely routes compared to the other, less-plausible routes is not derived by chance. The probabilities are calculated for mean annual temperature anomaly relative to 90 ka (T), mean annual precipitation relative to 90 ka (P), mean distance to nearest coast (d_{coast}), percentage of forest since 90 ka (veg), mean distance to the nearest main river (d_{river}), and mean ruggedness (rug). We accounted for the effect of the length on P_{rand} by normalising the environmental values along each route by its respective length (see details in *Methods*). Also displayed, the average **length** of both the most-likely and less-plausible routes across all continents along with the **mean (\pm standard deviation)** and **range** (calculated as the difference between the maximum and minimum values) of each environmental variable for all non-normalised trajectories.”

2. Page 5 Line 125: “We extracted the genetic dataset from a GenBank search query to find 67,643 human mitochondrial control-region sequences, cleaned of ancient DNA data and any non-Indigenous lineages.”: the number of samples (67,643) seems misleading as in the end 27,506 samples were used. Either mention both the numbers or mention the final sample number that was used to calculate the F_{st} .

Response #6: We have amended the main text accordingly, only mentioning the final sample number in the *expansion route* section (P5, L126) and keeping both numbers in the *Methods* for full transparency and reproducibility regarding our data-acquisition procedure (P13, L359–365).

“We gathered a total of 67,643 human mitochondrial control region sequences from Genebank (last accessed 28 May 2017, the last available update in the regions of interest). We excluded ancient DNA data because they are too scarce or even non-existent in most regions of the world to run short tandem-repeat analyses and obtain enough nuclear coverage to address these movements of early modern humans. We then retained 27,506 sequences with reliable geographical locations to compute the fixation index (F_{ST}), a proven and robust method to infer genetic distance between populations⁸⁹.”

7. I am a little bit confused about how the environmental variables are calculated. For example: Page 16, Line 421: mean annual temperature anomaly (at the estimated timing of arrival; Extended Data Fig. 4a) relative to the present. Page 19, Line: mean annual precipitation relative to 90 ka. Page 4, line 97: after they exited Africa over 65 ka. Is it relative to the present or it is relative to 90ka which is

used for the calculation? Is it indeed 90 ka which is used for the Out of Africa event, as 65 ka was mentioned also? The authors should follow one parameter throughout the paper to avoid confusion. Specially the parameter in this case should be 65ka as the authors are not concentrating on the coastal path of out of Africa migration.

Response #7: We apologise for the confusion and thank the reviewer for pointing out our mistake. Indeed, we calculated mean annual temperature anomaly and mean annual precipitation anomaly relative to 90 ka and not relative to the present as we had mentioned in the main text. We have now amended the text accordingly (P18, L477–480):

“... mean annual temperature anomaly (at the estimated timing of arrival; Extended Data Fig. 4a) relative to 90 ka, (iv) mean annual precipitation anomaly (at the estimated timing of arrival; Extended Data Fig. 4a) relative to 90 ka, (v)...”

Reviewer #5

1. With respect to Reviewer 1 (Comments 1,2 Responses 1,2): The authors admit that they ignore the southern route out of Africa. There are a couple of concerns here. The first is that the southern route has been the subject of numerous publications, including numerous high impact publications, over two decades. To ignore the southern route seems entirely implausible to me given that the authors aim to examine the "initial expansion of anatomically modern humans". Even if the authors believe the southern route played no role in the dispersal of our species, the complete absence of any data or analysis is a concern.

Response #8: This is a good point, and we should point out that we did not ignore the southern route because we did not think it played a role in the expansion of *Homo sapiens*. Instead, we omitted this consideration because of potential technical and methodological biases that would emerge by including the southern routes, thereby increasing the uncertainties of our results.

But we have now drafted a new section of the *Methods* (P20–22, L542–574), in which we have detailed that including the southern route would require using other types of data in addition to radiocarbon dates (i.e., luminescence and uranium-series methods) that would lead to two potential problems: (1) a less-reliable regional estimate of the timing of arrival because of the different nature of the datasets and magnitude of age uncertainties¹⁰⁹, and (2) including data before 50,000 years ago would increase the risk of misclassifying the association between *Homo sapiens* with Middle Palaeolithic industries (that could in fact be associated with Neanderthals, Response #10).

“We acknowledge that by focusing on the Eurasian ‘northern route’ of expansion via the Fertile Crescent, we cannot address the major debate about the relative roles of coastal expansion and environmental adaptation along the southern spread route in Eurasian (via Arabia and south Asia) ... Middle Palaeolithic is associated with Neanderthals¹¹², except in a few regional cases (e.g., Chatelperronian of northern Spain/France).”

As such, we have now reframed the context of our paper around the role of environmental drivers of initial **northern expansion** of anatomically modern humans to tighten the focus of the manuscript (see title and P2, L34). While we acknowledge the importance of the south Eurasian route of expansion, our results are critical because they combine archaeology, genetics, and climate modelling approaches for the first time, together revealing the role of environmental adaptations (including coastal changes such as in the Pacific Northwest and northern Europe) to facilitate/hinder the spread of modern humans across > 90% of both Eurasia and the Americas.

“Ultimately, focusing on the northern route of expansion resulted in excluding a total land area of ~ 8,980,723 km² (i.e., sum of the areas of each excluded country described in Supplementary Method 1), which represents ~ 9.5% of our entire study area (i.e., Eurasia + Americas = ~ 94,000,000 km²). We therefore argue that despite the importance of the south Eurasian route of expansion, our results still show the role of environmental adaptations (including coastal changes such as in the Pacific Northwest and northern Europe) to facilitate/hinder the spread of modern humans across > 90% of both Eurasia and the Americas.”

2. With respect to Reviewer 1 (Comment 3, Response 3): The second problem related to the above, is the fact that the authors ignore all sites and dates with luminescence and uranium series methods. I find this to be unreasonable. Indeed, I find the study methodologically flawed from the outset if it relies only on radiocarbon dates for examining human fossils and archaeological sites associated with anatomically modern humans (essentially the same point as Reviewer 2, Comment 1).

Response #9: We have now addressed this issue in the new section of the *Methods* (Response #8) in which we detailed (P21, L548–557) why including sites dated with luminescence and uranium-series methods would be technically challenging and would reduce the statistical power of our approach. While we acknowledge that our approach would benefit in further development to include various type of data, at this stage this is too far out of the scope of our manuscript because it would be of limited benefit for investigating the northern expansion of anatomically modern humans.

“First, many sites presumably associated with early modern humans in the southern Asia regions are primarily dated using optically stimulated luminescence and uranium-thorium dating methods. Given the differences in the dating process and the nature of the age uncertainties of these approaches (i.e., magnitude, source of variance, etc.)¹⁰⁸, integrating multiple types of data into a common modelling framework would be challenging and most likely decrease the robustness of our estimated timing of arrival. While we acknowledge that our method should be further expanded with additional data from all chronometric methods to investigate the southern Eurasia route, we argue that in the context of the northern expansion route of *Homo sapiens*, the small proportion of these additional data relative to the 5,977 reliable radiocarbon dates we used here is unlikely to alter our results.”

3. With respect to Reviewer 1 (Comment 5, Response 5): The assumption that the Middle Palaeolithic

is associated with Neanderthals is flawed. There are clear associations of *Homo sapiens* with Middle Palaeolithic technologies outside of Africa! Please remember that the majority of the evolution of *Homo sapiens* in Africa was associated with the Middle Stone Age, so any dispersals prior to the Later Stone Age would be associated with Middle Palaeolithic toolkits. There are a number of publications which demonstrate this point.

Response #10: We agree that there are several examples showing that *Homo sapiens* could be associated with Middle Palaeolithic technology^{110,111} such as: (i) in the Levant, where early *Homo sapiens* was present at the same time as the Middle Palaeolithic and (ii) in Europe, where the lack of association of *Homo sapiens* with Middle Palaeolithic or Mousterian toolkits is most likely because they arrived in visible numbers later after the early Upper Palaeolithic had started.

However, as we have now clarified in the new *Methods* section (P21, L558–5662; Response #8), using a dataset only based on radiocarbon dates for the analysis decreases the risk of *neanderthalis/sapiens* misclassification. Therefore, including data from before 50,000 years ago, such as sites dated with luminescence and uranium-series methods (adding even more uncertainty), risks associating *Homo sapiens* to Neanderthal sites. Whereas we are confident that the Middle Palaeolithic is associated with Neanderthals¹¹², except in a few regional cases.

“Second, these ages estimated using optically stimulated luminescence and uranium-thorium methods often describe the presence of anatomically modern humans older than 50,000 years ago, which challenge our assumption that the Middle Palaeolithic is associated with Neanderthals, whereas any material from the Upper Palaeolithic was associated with anatomically modern humans⁸⁷. We concede that, before 50 ka, *Homo sapiens* could be associated with Middle Palaeolithic industries^{110,111}, but the fact that our dataset is only based on radiocarbon dates means that we are confident that the Middle Palaeolithic is associated with Neanderthals¹¹², except in a few regional cases (e.g., Chatelperronian of northern Spain/France).”

One of our co-authors (TH) has written a paper that will be published soon showing that one of the Late Middle Palaeolithic industries of Europe (the Lincombian-Ranisian-Jerzmanowician of the north European region, dated 49–43 ka) is not associated with Neanderthals as previously suspected, but is instead firmly associated with *Homo sapiens*.

References cited in this response.

- 27 Araujo, B. B. A., Oliveira-Santos, L. G. R., Lima-Ribeiro, M. S., Diniz-Filho, J. A. F. & Fernandez, F. A. S. Bigger kill than chill: The uneven roles of humans and climate on late Quaternary megafaunal extinctions. *Quaternary International* 431, 216-222 (2017). <https://doi.org:10.1016/j.quaint.2015.10.045f>
- 35 Signor, P. W. & Lipps, J. H. in *Geological implications of large asteroids and comets on the Earth*. Geological Society of America Vol. 190 (ed L.T. Silver & P. H. Schultz) 291-296 (1982).
- 50 Fehren-Schmitz, L., Harkins, K. M. & Llamas, B. A paleogenetic perspective on the early population history of the high altitude Andes. *Quaternary International* 461, 25-33 (2017). <https://doi.org:10.1016/j.quaint.2017.01.003>

- 66 Zhang, X. L. et al. The earliest human occupation of the high-altitude Tibetan Plateau 40 thousand to 30 thousand years ago. *Science* 362, 1049-1051 (2018).
<https://doi.org/10.1126/science.aat8824>
- 75 Raup, D. M. Taxonomic Diversity during the Phanerozoic. *Science* 177, 1065-1071 (1972).
<https://doi.org/10.1126/science.177.4054.1065>
- 76 Wang, S. C. & Marshall, C. R. Estimating times of extinction in the fossil record. *Biol. Lett.* 12 (2016). <https://doi.org/10.1098/rsbl.2015.0989>
- 77 Bartlett, L. J. et al. Robustness despite uncertainty: regional climate data reveal the dominant role of humans in explaining global extinctions of Late Quaternary megafauna. *Ecography* 39, 152-161 (2016). <https://doi.org/10.1111/ecog.01566>
- 78 Sandom, C., Faurby, S., Sandel, B. & Svenning, J.-C. Global late Quaternary megafauna extinctions linked to humans, not climate change. *Proc. R. Soc. B* 281, 20133254 (2014).
<https://doi.org/10.1098/rspb.2013.3254>
- 79 Emery-Wetherell, M. M., McHorse, B. K. & Byrd Davis, E. Spatially explicit analysis sheds new light on the Pleistocene megafaunal extinction in North America. *Paleobiology*, 1-14 (2017). <https://doi.org/10.1017/pab.2017.15>
- 80 Benton, M. J. Palaeodiversity and formation counts: redundancy or bias? *Palaeontology* 58, 1003-1029 (2015). <https://doi.org/10.1111/pala.12191>
- 87 Higham, T. et al. The timing and spatiotemporal patterning of Neanderthal disappearance. *Nature* 512, 306-309 (2014). <https://doi.org/10.1038/nature13621>
- 89 Jorde, L. B. et al. The distribution of human genetic diversity: a comparison of mitochondrial, autosomal, and Y-chromosome data. *Am. J. Hum. Genet.* 66, 979-988 (2000).
<https://doi.org/10.1086/302825>
- 108 Hoogakker, B. A. A. et al. Terrestrial biosphere changes over the last 120 kyr. *Climate of the Past* 12, 51-73 (2016). <https://doi.org/10.5194/cp-12-51-2016>
- 109 Zipkin, E. F. et al. Addressing data integration challenges to link ecological processes across scales. *Front. Ecol. Environ.* 19, 30-38 (2021). <https://doi.org/10.1002/fee.2290>
- 110 Devièse, T. et al. Reevaluating the timing of Neanderthal disappearance in Northwest Europe. *Proc. Natl. Acad. Sci.* 118, e2022466118 (2021).
<https://doi.org/10.1073/pnas.2022466118>
- 111 Been, E. et al. The first Neanderthal remains from an open-air Middle Palaeolithic site in the Levant. *Sci. Rep.* 7, 2958 (2017). <https://doi.org/10.1038/s41598-017-03025-z>
- 112 Semal, P. et al. New data on the late Neandertals: Direct dating of the Belgian Spy fossils. *Am. J. Phys. Anthropol.* 138, 421-428 (2009).
<https://doi.org/10.1002/ajpa.20954>

Reviewers' Comments:

Reviewer #1:

Remarks to the Author:

I am pleased that the authors have put much effort to clarify the manuscript. Some are not easy to address for the lack of original data. There is a remind to the third response to my previous comments. The Nwya Devu seems not to be the strong evidence to support the dispersal across Himalaya. The reason is simple that the site is the only site on the Tibet with such an early age, and modern humans(hypothetically) could have come from many directions not necessarily across Himalaya. And the authors of the Nwya Devu paper seem to prefer a north to south route from northwestern China and Siberian Altai.

Reviewer #4:

Remarks to the Author:

Authors have improved the manuscript but it still needs some clarification before accepted:

1. I agree with the authors that in Table 1 sd are so high that the differences in mean are negligible. But if that is true most of the Figure 2 and their corresponding discussion in the results section needs an update. It would be great if the authors add some p value tests for every trajectories between most likely vs. less plausible routes and put a star (or any indication) for those significant differences on the top of the graphs (after the confidence interval). Authors should only discuss those in the results section which are significantly different (for example, from Figure 2a Africa to Berengia and Berengia to Mexico look significant only).
2. It should be clarified why 90ka was used in Table 1 instead of 65ka as the authors are only concentrating on northern routes.

Environmental conditions shaped the patterns of initial northern expansion of anatomically modern humans

Reviewer #1

1. I am pleased that the authors have put much effort to clarify the manuscript. Some are not easy to address for the lack of original data. There is a remind to the third response to my previous comments. The Nwya Devu seems not to be the strong evidence to support the dispersal across Himalaya. The reason is simple that the site is the only site on the Tibet with such an early age, and modern humans(hypothetically) could have come from many directions not necessarily across Himalaya. And the authors of the Nwya Devu paper seem to prefer a north to south route from northwestern China and Siberian Altai.

Response #1: We have addressed this feedback (P9-10, L243-246) by now conceding that although our findings are statistically sound and prompt intriguing questions, additional evidence is required to substantiate a major migration event through the Himalayas.

“Although further evidence is necessary to support the hypothesis that humans primarily traversed the Himalayas rather than taking alternative paths, these findings indicate that rugged terrain might not have posed insurmountable obstacles to their migration^{2,54}.”

Reviewer #4

2. I agree with the authors that in Table 1 sd are so high that the differences in mean are negligible. But if that is true most of the Figure 2 and their corresponding discussion in the results section needs an update. It would be great if the authors add some p value tests for every trajectories between most likely vs. less plausible routes and put a star (or any indication) for those significant differences on the top of the graphs (after the confidence interval). Authors should only discuss those in the results section which are significantly different (for example, from Figure 2a Africa to Berengia and Berengia to Mexico look significant only).

Response #2: We have now constructed a regional randomisation test to indicate for each region and each environmental variable the probability (denoted $\text{Pr}(R)_{\text{rand}}$ in the text) that the difference in any environmental variable between the main trajectories and alternate trajectories could arise by chance. We described this approach on P20-21, L528-551:

“... we constructed a global and a regional the following randomisation test, [...] that the median pairwise difference in any environmental variable between the main trajectories and alternate trajectories could arise by chance.”

We incorporated the test results into an inset table in each panel of Figure 2 (we have also amended its caption on P36, L946-949) and revised the Discussion (P9-11, L219-282) to limit our argument to the cases where $\text{Pr}(R)_{\text{rand}}$ is low.

“However, these environmental variables are intertwined, [...] led a westward expansion of Amazonian populations into the Andes despite colder climate conditions

(drier and more densely forested; Fig. 1, 2a, and Extended Data Fig. 6b, d) compared to routes from the Pacific littoral or a trans-Altiplanic entry from the north⁵⁰.”

3. *It should be clarified why 90ka was used in Table 1 instead of 65ka as the authors are only concentrating on northern routes.*

Response #3: We have now clarified the caption of Table 1 (P35, L917-919) accordingly:

“We chose 90 ka as an arbitrary baseline because it falls within the timeframe when anatomically modern humans left Africa permanently¹⁰.”

References cited in this response:

- 2 Timmermann, A. & Friedrich, T. Late Pleistocene climate drivers of early human migration. *Nature* **538**, 92-95 (2016). <https://doi.org:10.1038/nature19365>
- 10 Grove, M. *et al.* Climatic variability, plasticity, and dispersal: A case study from Lake Tana, Ethiopia. *J. Hum. Evol.* **87**, 32-47 (2015). <https://doi.org:10.1016/j.jhevol.2015.07.007>
- 50 Fehren-Schmitz, L., Harkins, K. M. & Llamas, B. A paleogenetic perspective on the early population history of the high altitude Andes. *Quaternary International* **461**, 25-33 (2017). <https://doi.org:10.1016/j.quaint.2017.01.003>
- 54 Bradshaw, C. J. A. *et al.* Stochastic models support rapid peopling of Late Pleistocene Sahul. *Nat. Commun.* **12**, 2440 (2021). <https://doi.org:10.1038/s41467-021-21551-3>

Reviewers' Comments:

Reviewer #4:

Remarks to the Author:

I am pleased that the authors clarified all my queries properly. I have no further comments.